# Inter-decadal climate variability induces differential ice response along Pacific-facing West Antarctica

Frazer D. W. Christie ⓘ[1,2] ✉, Eric J. Steig ⓘ[3], Noel Gourmelen ⓘ[2], Simon F. B. Tett ⓘ[2] & Robert G. Bingham ⓘ[2]

West Antarctica has experienced dramatic ice losses contributing to global sea-level rise in recent decades, particularly from Pine Island and Thwaites glaciers. Although these ice losses manifest an ongoing Marine Ice Sheet Instability, projections of their future rate are confounded by limited observations along West Antarctica's coastal perimeter with respect to how the pace of retreat can be modulated by variations in climate forcing. Here, we derive a comprehensive, 12-year record of glacier retreat around West Antarctica's Pacific-facing margin and compare this dataset to contemporaneous estimates of ice flow, mass loss, the state of the Southern Ocean and the atmosphere. Between 2003 and 2015, rates of glacier retreat and acceleration were extensive along the Bellingshausen Sea coastline, but slowed along the Amundsen Sea. We attribute this to an interdecadal suppression of westerly winds in the Amundsen Sea, which reduced warm water inflow to the Amundsen Sea Embayment. Our results provide direct observations that the pace, magnitude and extent of ice destabilization around West Antarctica vary by location, with the Amundsen Sea response most sensitive to interdecadal atmosphere-ocean variability. Thus, model projections accounting for regionally resolved ice-ocean-atmosphere interactions will be important for predicting accurately the short-term evolution of the Antarctic Ice Sheet.

Three decades of continuous Earth observation have shown significant ice loss from West Antarctica in response to ocean-driven ice-shelf melt and consequent thinning and recession of inland ice[1]. Marked and accelerating losses from Pine Island and Thwaites glaciers into the Amundsen Sea, especially, may signify the onset of an irreversible ice-sheet collapse, with important implications for global sea-level rise[2–4]. These phenomena have been ascribed both to event-based atmospheric forcing, such as the possible 1940s El-Niño-triggered ungrounding of Pine Island Glacier Ice Shelf[5], and a long-term trend in forcing over the 20th Century[6]. Previous modelling of Thwaites Glacier's evolution has signalled that "bed topography controls the pattern of grounding-line retreat, while oceanic thermal forcing

impacts the rate of grounding-line retreat"[7], but to date, observations of this relationship are lacking beyond the Amundsen Sea Embayment. Several studies have shown that much of West Antarctica's Pacific coast has experienced ice loss over the satellite era, evinced by ice-flow acceleration[8,9], ice thinning[10,11] and retreat of the grounding line[9,12,13]. However, these previous studies have not been performed over consistent timescales all along West Antarctica's Pacific-facing margin, limiting our ability to evaluate how atmospheric and oceanic forcing regulate ice-sheet losses.

Here, we present a comprehensive analysis of glaciological change across the Pacific margin of West Antarctica for the period 2003-2015. We quantify changes in grounding-line position along 94%

[1]Scott Polar Research Institute, University of Cambridge, Cambridge CB2 1ER, UK. [2]School of GeoSciences, University of Edinburgh, Edinburgh EH8 9XP, UK. [3]Department of Earth & Space Sciences, University of Washington, Seattle, WA 98195-1310, USA. ✉e-mail: fc475@cam.ac.uk

of the ~3500-km coastal margin, and compare these records to contemporaneous changes in ice-surface velocity and ice-sheet mass loss. We then assess the correspondence of the observed glaciological changes with outputs from a suite of atmosphere and ocean reanalysis datasets.

## Results

### Glaciological change

We determined rates of grounding-line migration along the West Antarctic margin using optical satellite-based techniques, supplemented in fast-flowing regions with synthetic aperture radar-based (SAR) observations, over the periods 2003-2008 and 2010-2015 (Methods; Supplementary Figs. 1–3). These periods coincide with the greatest availability of grounding-line information along the coastline[3,12–18], which we have augmented to derive a complete record of grounding-line migration rate change along the entire Pacific

margin of West Antarctica (Fig. 1; Supplementary Data 1 and 2). We also calculated near-contemporaneous change in ice-flow acceleration between the two periods using several ice-velocity mosaics derived from satellite imagery acquired between 1999 and 2015 (Methods; Fig. 1 and Supplementary Fig. 4; Supplementary Table 1). Our results show clear catchment-wide and regional trends of change across the Amundsen, Bellingshausen and Ross Sea coastlines (Fig. 1). Along the Bellingshausen Sector, grounding-line retreat rates increased along the majority of the coastline through time, in conjunction with notable increases in ice-flow acceleration near the grounding zone of the fastest-flowing outlet glaciers (Fig. 1 and Supplementary Figs. 2 and 4). The largest increase in grounding-line recession (218 m yr⁻¹) occurred within Eltanin Bay (Fig. 1 and Supplementary Figs. 2, 4 and 5), where no major ice shelf has existed over the observational era. Eltanin Bay is thought to have been characterized by ocean-driven ice-dynamical imbalance over at least the past 40 years[10,12,19]. Along the entire

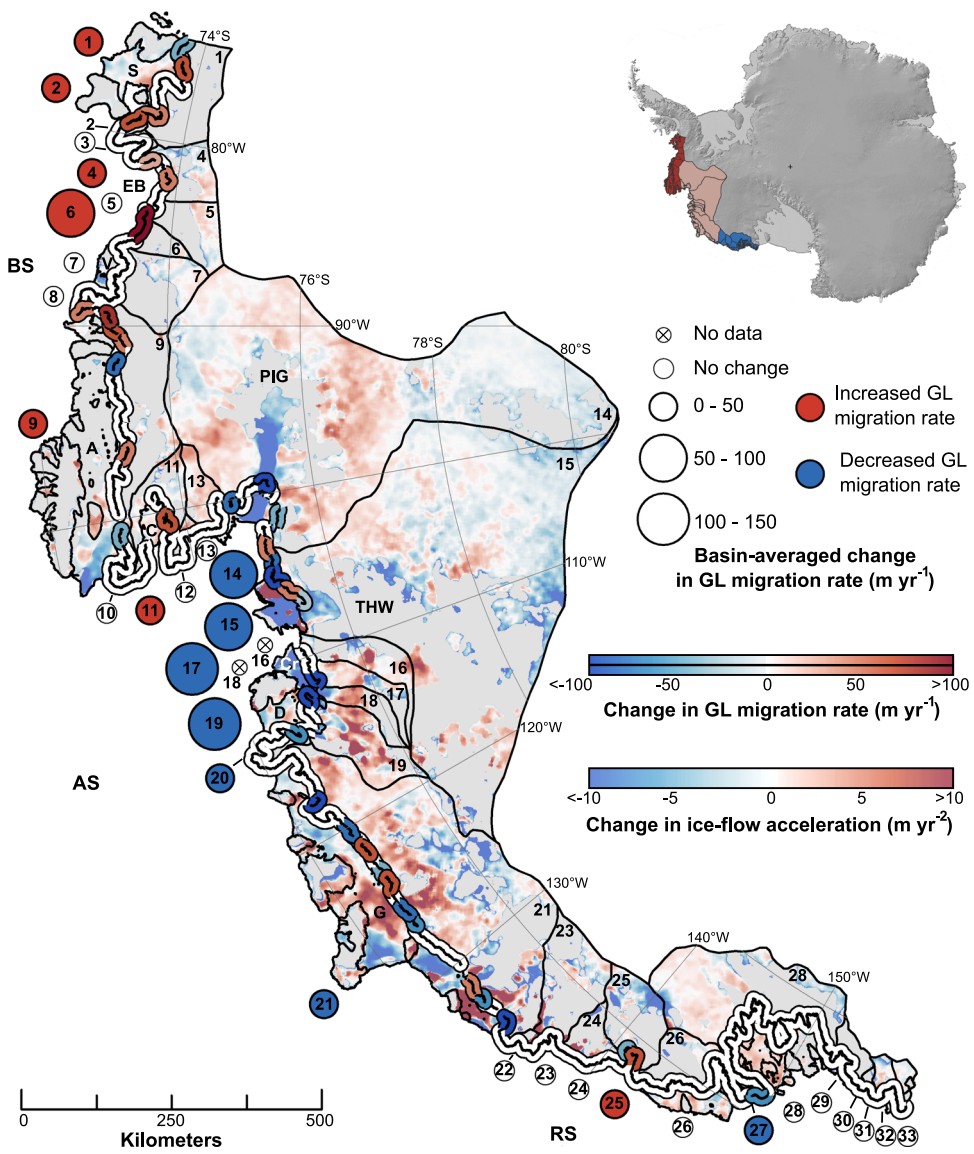

**Fig. 1 | Glaciological change across West Antarctica's Pacific-facing margin, 2003-2015.** The thick curve bounding the coastline shows net change in grounding-line migration rate (m yr⁻¹) over the observational period (c. 2010-2015 minus c. 2003-2008), binned into 30 km segments along the grounding line. Also shown is catchment-averaged change in grounding-line migration rate for each glacial basin[83] along the coastline (numbered circles). Data are superimposed over near-contemporaneous change in ice-flow acceleration (m yr⁻²) (Methods). AS

denotes Amundsen Sector, BS Bellingshausen Sector, RS Ross Sector, S Stange Ice Shelf, EB Eltanin Bay, V Venable Ice Shelf, A Abbot Ice Shelf, C Cosgrove Ice Shelf, PIG Pine Island Glacier, THW Thwaites Glacier; Cr Crosson Ice Shelf, D Dotson Ice Shelf and G Getz Ice Shelf. Note that the glacial basins shown are from MEaSUREs[83] but for ease of reference we have numbered them 1-33 from east to west. Inset shows the location of the study domain, partitioned into the Bellingshausen (red), Amundsen (pink) and Ross Sea sectors (blue). Inset background is REMA DEM[84].

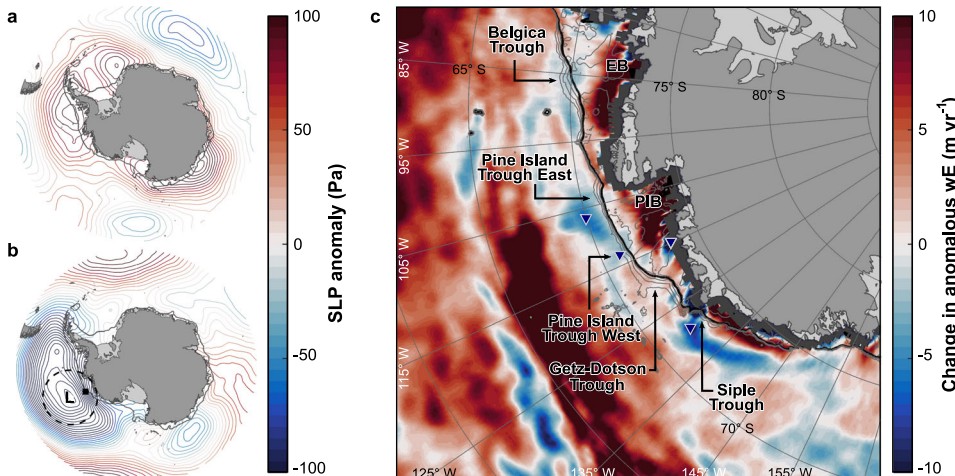

**Fig. 2 | Southern Ocean atmosphere-ocean change, 2003-2015. a** ERA5-derived mean sea-level pressure (*SLP*, Pa) anomaly, 2003-2008, relative to the period 1979-2015. **b** same as **a**, but for 2010-2015. Dashed ring denotes 1000 km radius from the central pressure location of the Amundsen Sea Low (*L*). Note the sharp gradient in *SLP* abeam the Amundsen Sector (solid square and surrounds). **c** change in ERA5-derived anomalous Ekman vertical velocity (*wE*; m yr⁻¹; negative denotes less upwelling) between the two periods (i.e. $\overline{wE}_{2010-2015} - \overline{wE}_{2003-2008}$; see also Supplementary Fig. 7). Data have been upsampled by a factor of two for mapping purposes. Dark grey denotes no data. Contours show sea-floor depth (500 m increments; grey lines) on and near the continental shelf [85]. EB denotes Eltanin Bay, PIB Pine Island Bay. Note the change towards reduced upwelling (negative *wE*) near the entrances of the Pine Island East, West and Siple troughs (blue triangles). In all plots, the black contour denotes the continental-shelf break limits (1000 m depth[85]).

Bellingshausen coastline, net grounding-line retreat during 2010-2015 was, on average, 13 m yr⁻¹ faster than during 2003-2008. Between these periods, both Stange and Abbot ice shelves and their feeder glaciers also experienced enhanced rates of thinning[10,20–22], while Venable Ice Shelf continued to thin at a high rate[20].

Our results show that, in contrast to the Bellingshausen coastline, rates of grounding-line retreat slowed pervasively along the Amundsen Sea Embayment and Getz Ice Shelf between 2003 and 2008 and 2010 and 2015. The most notable slowdowns occurred at the Pine Island (56 m yr⁻¹), Thwaites (84 m yr⁻¹), Pope (126 m yr⁻¹) and Smith/Kohler (107 m yr⁻¹) basins, while along Getz Ice Shelf, grounding-line retreat slowed by 24 m yr⁻¹ on average (Fig. 1 and Supplementary Fig. 2; Supplementary Data 2). Whereas localised increases in ice-flow acceleration were detected along parts of the coastline (including, for example, stretches of central Getz Ice Shelf in line with previous research[8,23]), the observed slowdown in grounding-line retreat was largely associated with a decrease in the pace of ice-flow acceleration, including at and inland of the Pine Island, Thwaites and Smith glaciers' grounding lines (Fig. 1). Contemporaneously, ice shelves along this stretch of the coastline experienced reduced rates of thinning by up to 51% relative to the 25-year altimetry record[18,22]. Negligible change in grounding-line migration rate was detected between Abbot Ice Shelf and Pine Island Glacier, except for a local unpinning on the southern flank of Cosgrove Ice Shelf (Fig. 1). Collectively, our observations reveal a clear slowdown in the rate of ice-dynamical imbalance between 2003-2008 and 2010-2015 across the Amundsen Sector.

At the Ross-Sea-facing margin, little change in grounding-line migration rate was observed over either satellite period (Fig. 1 and Supplementary Fig. 2; Supplementary Data 2). Together with more limited change in ice-flow acceleration and ice thinning rates relative to elsewhere in the domain[1,9,18,20–22], these observations imply the region to be divorced from atmosphere-ocean-driven dynamic imbalance.

## Atmospheric variability

We show that the contrasting behaviour of the Bellingshausen and Amundsen ice margins between 2003 and 2015 can be attributed to inter-decadal atmospheric variability. Between the earlier (2003–2008) and later (2010–2015) periods analysed, ECMWF ERA5 reanalysis data (Methods) reveal that there was a significant decrease in mean sea-level pressure (*SLP*) over the Southern Ocean bordering West Antarctica, characteristic of a deepening of the Amundsen Sea Low[24] (Fig. 2a, b). This brought an intensification of near-shore easterly winds, most pronounced abeam the Amundsen Sea coastline, as exhibited by a steep meridional gradient in mean *SLP* towards the coastline (Fig. 2b).

Previous work[5,6,25] has demonstrated that westerly winds at the continental-shelf break promote the upwelling and ingress of relatively warm circumpolar deep water (CDW) to the coast, and hence the intensification of opposing easterlies that we observe implies suppressed CDW access to the Amundsen margin in 2010-2015. No region-wide CDW records exist spanning 2003-2015, but Ekman vertical velocity (as calculated from a variety of global reanalyses products; Methods) reduced significantly (implying less upwelling) near the entrance of Pine Island Trough East (~−5 m yr⁻¹; Fig. 2c and Supplementary Figs. 6 and 7), the primary routeway of CDW to the Amundsen Sea Embayment[4,9,24,25]. More muted reductions in Ekman Vertical Velocity (~−1 m yr⁻¹) also occurred near the mouth of the tributary Pine Island Trough West.

East of the Amundsen Sea Embayment, Ekman velocity changed little from the high rates near the continental-shelf break of the Bellingshausen Sea, with only minor amounts of reduced upwelling (~−1 m yr⁻¹) detected around the entrance of the deep continental-shelf-bisecting Belgica Trough (Fig. 2c and Supplementary Figs. 5–7). This is consistent with the near-sustained provision and shoaling of CDW onto the Bellingshausen continental shelf through time[12,19,26,27]. West of the Amundsen Sea Embayment, however, a significant reduction in near-shore Ekman velocity (Fig. 2c and Supplementary Figs. 6 and 7; Methods) implies a suppression of CDW inflow to the Dotson and Getz sub-ice-shelf cavities relative to 2003-2008. Reductions in Ekman velocity at the entrance of Siple Trough, another key pathway for CDW ingress onto the continental shelf[23], will have further inhibited CDW circulation beneath Getz Ice Shelf (Fig. 2c).

## Oceanic variability

The wind-driven changes in ocean conditions we infer from the atmospheric reanalyses (Fig. 2 and Supplementary Figs. 6 and 7) are corroborated directly by oceanographic data. Four different ocean products (Methods), all of which are constrained by in-situ oceanographic observations, reveal changes in the vertical hydrography

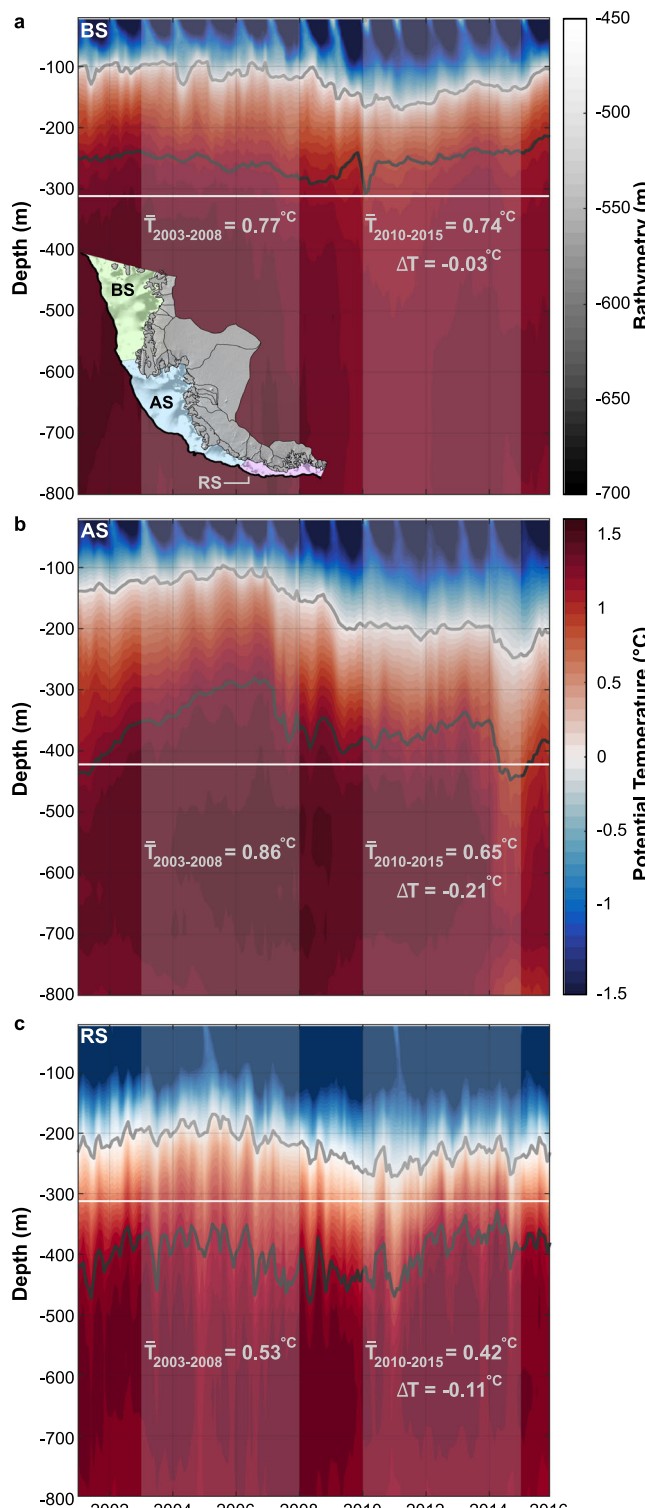

**Fig. 3 | Vertical hydrography offshore from Pacific-facing West Antarctica, 2003-2015.** CMEMS GREP ocean potential temperature (°C) derived from all grid cells on the (**a**) Bellingshausen (BS), (**b**) Amundsen (AS) and (**c**) Ross Sea (RS) continental-shelf regions (green, blue and pink in inset of panel *a*, respectively). Light grey contours denote the 0 °C thermocline; dark grey, the upper limit of circumpolar deep water at 34.62 PSU (Methods). White lines denote the mean elevation of the grounding line across each sector[85]; translucent grey patches, the periods 2003–2008 and 2010–2015. In each panel, mean ocean potential temperature >0 °C at and above the mean depth of the grounding line during each period is also shown, as is the change in temperature between each period (ΔT). In inset, greyscale shading reveals the deep, cross-shelf cutting troughs[85] connecting the continental-shelf break (thick black line as in Fig. 2c) to the ice-sheet margin. Inset background is REMA DEM[84].

the mean depth of the grounding line. This reveals that the draft of CDW across the Amundsen Sea continental shelf diminished through time. In-situ observations revealing similar trends of oceanic cooling from ~2010 at Pine Island Bay, Dotson and Getz ice shelves align with these findings[25,28–30].

In the Ross Sea, the upper limit of CDW never shoaled above the mean depth of the grounding line (Fig. 3). The few available observations from the continental shelf fringing this coastline[26] reinforce the interpretation of only limited, intermittent CDW presence in recent decades. A deeper surface water layer is also observed throughout this sector, consistent with the presence of fresher on-shelf waters and the advection of cold, low-salinity glacial meltwater derived from the Amundsen Sea[31]. We therefore interpret the Ross Sea Sector as having been impervious to atmosphere-ocean variability over our observational timeframe.

## Discussion

Over our period of observation, the Bellingshausen Sea has been steadily flooded by relatively warm circumpolar deep water (CDW; Fig. 3), which has reached much of the ice-sheet margin and facilitated the dynamic thinning detected by satellites (Fig. 1). This is exemplified on the coastline between the Stange and Abbot ice shelves, where rates of grounding-line retreat, outlet-glacier ice flow (Fig. 1 and Supplementary Fig. 2) and ice thinning[1,10,12,20] have accelerated. This stretch of coastline corresponds to where the ice margin overlies the CDW-laden tributaries of the deeply incised Belgica Trough[12,19], which lies in direct contact with the Antarctic Circumpolar Current at the continental-shelf break[12,26,27]. The ice-atmosphere-ocean coupling is maximised in Eltanin Bay, where ice residing immediately upstream of Belgica Trough's central trunk exhibited the largest increase in grounding-line retreat (Figs. 1, 2c and Supplementary Fig. 5). This indicates conditions in the Bellingshausen Sea conducive both to the sustained transmission of CDW from the continental-shelf break and its steady delivery to the ice margin.

Along the Amundsen Sea region, there is clear inter-decadal variability in the volume of CDW transported across the continental shelf. The strong reduction in CDW in the 2010-2015 period compared with 2003-2008 (Fig. 3) coincided with an unambiguous reduction in the rate of grounding-line retreat in this area (Fig. 1 and Supplementary Fig. 2). This was associated with a reduction in the pace of ice-flow acceleration along the grounding line of most outlet glaciers (Fig. 1) and, by implication, a slowing in the rate of ice-dynamical imbalance through time. Recent observations of reduced ice-thinning rates[18,22,32,33] and a stabilisation in mass balance trends[1,34] from *c.* 2010 in the Amundsen Sector (Methods; Supplementary Fig. 8) support these observations. The reduced volume of CDW in the eastern Amundsen Sea (Fig. 3) can be attributed to its suppressed delivery to the deep-bedded entrance of the eastern and western Pine Island troughs, especially, in turn a consequence of reduced upwelling at the continental-shelf break (Fig. 2c). Changes in near-shore upwelling also explain the net slowdown in grounding-line migration rate and

between 2003-2015 (Fig. 3) of the same sign and spatial pattern indicated by our Ekman velocity calculations. The ensemble mean of these four products shows that on the Bellingshausen Sea continental shelf, the upper boundary of CDW varied little with time and was persistently above the mean depth of the ice-sheet grounding line (Fig. 3), which is consistent with independent ocean observations[26,27]. In contrast, in the Amundsen Sea encompassing the continental-shelf waters seaward of both the Amundsen Sea Embayment and Getz Ice Shelf regions (Fig. 3), the upper bound of CDW deepened by tens to hundreds of metres over the same period towards

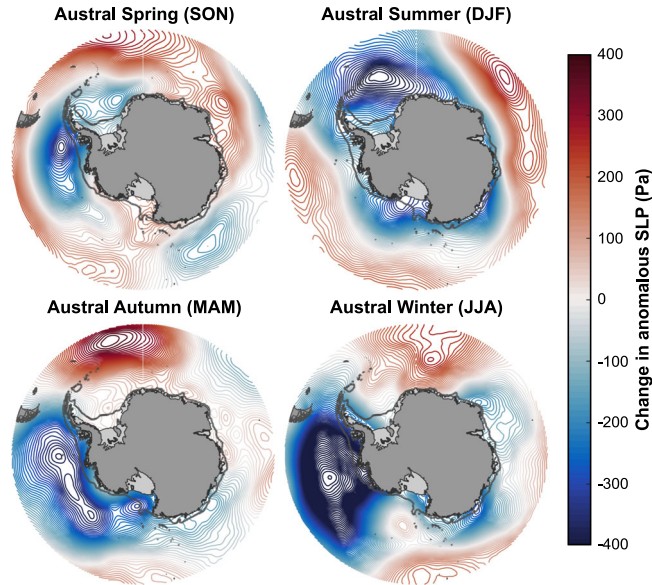

**Fig. 4 | Seasonal Southern Ocean atmospheric change, 2003–2015.** Difference in anomalous seasonal sea-level pressure (*SLP*) between the periods 2003-2008 and 2010-2015.

reduction in the pace of ice-flow acceleration at Dotson Ice Shelf (Fig. 1 and Supplementary Fig. 2), which reduced ice-shelf-averaged thinning[22] and facilitated a 2-km re-advance of Kohler Glacier from *c.* 2011[13,29]. Similar mechanisms reduced upwelling and limited CDW ingress beneath Getz Ice Shelf [30], consistent with the net reduction in grounding-line retreat rate we observe along this stretch of the coastline (Fig. 1). Collectively, our observations show that the rate of ice response (grounding-line retreat and ice-flow acceleration change) from the Amundsen Sea Embayment to the western Getz Ice Shelf is in-phase with atmosphere-ocean forcing over inter-decadal timescales. The pervasiveness of this relationship along a margin transected by multiple glaciers, with bed slopes ranging from reverse to prograde[9], signals the primary influence of the inter-decadal forcing on rates of glaciological change.

The close correspondence between glacier behaviour and atmosphere-ocean variability we observe (Figs. 1–3) clearly implicates wind-driven oceanic influence as the major forcing mechanism. Notably, the Amundsen Sector lay within ~1000 km from the predominant deepening pressure centre of the Amundsen Sea Low during our observational period, equivalent to the synoptic-scale length of atmospheric deformation at high latitudes[35]; while most of the Bellingshausen Sector lay beyond (Fig. 2b). Several studies have implicated the depletion of stratospheric ozone and related changes to the Southern Annular Mode (SAM), an index for the poleward contraction and strength of the westerly winds encircling Antarctica, in the deepening of the Amundsen Sea Low[36,37]. In general, when SAM is in positive polarity (SAM +), sea-level pressure in the region of the Amundsen Sea Low is lower[24]. Increasingly positive SAM + conditions between 2003 and 2015 (Supplementary Fig. 9, Methods) may therefore partly explain the intensification of the Amundsen Sea Low we find between 2003 and 2015 (Fig. 2b), although, apart from austral summertime, seasonal partitioning of this phenomenon (Fig. 4; Methods) reveals zonally asymmetric *SLP* anomalies around Antarctica uncharacteristic of SAM's typical spatial signature[5,38]. These seasonal patterns suggest that an additional large-scale forcing mechanism, bolstered by ongoing SAM + conditions, dominated West Antarctic ice-ocean-atmosphere interaction over the timescales we consider.

Beyond SAM, modelling of atmospheric and oceanic forcing in the Amundsen Sector has linked the wind-driven incursion of CDW to far-field perturbations in atmospheric circulation, with the principal teleconnection being the El Niño-Southern Oscillation (ENSO)[5,6,25,39,40]. Consistent with the seasonal *SLP* observations shown in Fig. 4, this teleconnection is most prevalent during non-summertime months[5,40]. Over our period of observation, a distinct transition from relatively sustained El Niño- to strong, prolonged La Niña-like conditions from *c.*2010 (Supplementary Fig. 9) can therefore explain the anomalous deepening of the Amundsen Sea Low we detect (Figs. 2b and 4; cf. refs. 21,24). Critically, this transition was also synchronous with the timing of reduced grounding-line retreat rates and ice-flow acceleration we observe along the Amundsen Sector (Fig. 1 and Supplementary Fig. 9). Our findings therefore suggest that the wider Amundsen Sea glacier margin was strongly affected by ENSO-dominated changes in the Amundsen Sea Low between 2003 and 2015, embodied by fluctuations in dynamic thinning. In contrast, the Bellingshausen Sea was less affected by such atmospheric variability, owing to both its relative remoteness from the long-term central pressure location of the Amundsen Sea Low and the more pervasive flow of CDW onto the continental shelf in this region (Figs. 2 and 3). Our observations therefore build upon previous research largely limited to the Amundsen Sea Embayment that has also shown an intimate coupling between inter-decadal atmosphere-ocean variability and ice-sheet change[21,24,25,29,41]. They suggest that in fact the coupling is particularly strong in the Amundsen Sea Embayment compared with elsewhere. This aligns with a previous assertion deduced from coarse-resolution modelling that interannual ocean variability in the Bellingshausen Sea is weaker than in the neighbouring Amundsen Sea[42].

Our results provide spatially extensive and systematic observations that the glaciological evolution of West Antarctica's Pacific-facing margin between 2003 and 2015 was controlled by the variability with which CDW accessed glacier fronts. Ice margins fringing the Ross Sea changed little reflecting a general barrier to CDW ingress; ice response along the Amundsen Sea was highly sensitive to CDW fluctuations driven by inter-decadal variability in the position of the Amundsen Sea Low; while grounding lines meeting the Bellingshausen Sea retreated progressively in response to pervasive and persistent CDW forcing less susceptible to atmospheric forcing. Our data are now available to underpin observationally-constrained modelling experiments to elucidate and predict short-term atmosphere-ocean-induced variable rates of West Antarctic ice loss superimposed onto the longer-term projected retreat trend[2,7,43,44].

## Methods

An analysis of glaciological, atmospheric and oceanic changes along the entire Pacific-facing margin of West Antarctica must be underpinned by systematic records. Prior to this study, we generated grounding-line location records between 2003 and 2015 along West Antarctica's Marie Byrd Land coast using optical satellite-based techniques[18,45], hence our focus on this time period. Other studies have mapped different proxies for the grounding line since 2015 using independent techniques, including altimetry and machine learning applied to InSAR observations[46,47]. However, reconciling grounding-line location proxies identified by different techniques is subject to bias; hence we chose not to extend our study period beyond 2015.

### Grounding-line migration

While no satellite-based technique can directly image the grounding line, various proxies can be used depending on the sensors available for a given timeframe[48]. For this study, we primarily used optical satellite imagery due to its superior temporal and spatial coverage over the full period 2003–2015 compared with observations from other sensors.

For our optical-imagery analysis, we tracked the ice-sheet break-in-slope across the ice-sheet-shelf boundary (otherwise referred to as the "inflexion point"; Point $I_b$[48]), building upon previous studies that

have shown that migration of $I_b$ over periods of years reliably captures grounding-line migration[12,18,49]. We also tracked the seaward limit of grounded glaciers draining directly to the ocean (i.e. those without an ice shelf). We used imagery acquired by the Enhanced Thematic Mapper (ETM + ; 2003, 2008, 2010) and Operational Land Imager (OLI; 2015) sensors on-board Landsat 7 and Landsat 8, respectively, in our analysis (Supplementary Fig. 1). This imagery is publicly available, and was acquired free of charge from the United States Geological Survey's 'Earth Explorer' data archive (https://earthexplorer.usgs.gov/). Grounding lines were mapped using all available cloud-free satellite imagery acquired during the height of austral summertime (Jan/Feb ± 2 months depending on cloud cover; cf. Extended Data 1), and represent the mean summertime grounding-line location for each year. Epochal rates of net grounding-line migration (2003-2008, 2010-2015; m yr[−1]) binned into 30 km segments along the coast were then calculated using recently documented techniques[45] (Supplementary Fig. 2 and Supplementary Data 2), and subsequently differenced to calculate the change in rate of grounding-line migration, $\Delta GL_{rate}$, between the two epochs (Fig. 1; Supplementary Data 2).

Positional uncertainties associated with our Landsat-derived grounding lines, calculated on a per-segment basis and derived as a function of grounded-ice boundary classification and sensor-specific geometric errors, are estimated to range between 32 and 500 m along the coast (Supplementary Data 2; after refs. 12, 18, 45, 49). These values were then summed in quadrature to obtain the absolute per-segment positional uncertainty associated with each epoch, $U_{GLR\_epoch}$ (Supplementary Fig. 3). Such uncertainty propagation is required due to the use of successive (multi-) sensor observations through time (i.e. combined Landsat 7-Landsat 7 (for years 2003 and 2008) or Landsat 7-Landsat 8 (for years 2010 and 2015) observations[18,45,49]), and these derived values were subsequently used to ascertain the overall between-epoch uncertainty, $U_{\Delta GLR}$, associated with our $\Delta GL_{rate}$ calculations (Fig. 1; Supplementary Fig. 3; Supplementary Data 2). To generate Fig. 1, we considered all $\Delta GL_{rate}$ values falling within $U_{\Delta GLR}$ to represent negligible change within satellite error bounds, and set all corresponding values to zero (Supplementary Fig. 3; Supplementary Data 2).

In fast-flowing regions of the Amundsen Sector, where grounding-line detection from optical imaging can be prone to higher positional uncertainty[50], we supplemented our optical-based analyses with recent, near-contemporaneous synthetic aperture radar (SAR) records of the ice-sheet-shelf limit of tidal flexure, Point $F^{3,12}$ (Supplementary Data 2). These records were obtained from high-precision, double-differenced interferometric synthetic aperture radar (DInSAR)-based analyses of repeat-pass ERS-1/2[16,17] (for years 2000, 2011), Sentinel-1a/b[13,16] (2016) and COSMO-Skymed[14] (2016, 2017) observations, and from coarser-resolution speckle-tracking applied to TerraSAR-X images[15] (2009, 2014) where no publicly available DInSAR observations exist (Supplementary Fig. 1). Along a small (10 km) section of Thwaites Glacier's centreline (Supplementary Fig. 1), 1996 ERS-1/2 DInSAR observations[17] were also used in lieu of any later grounding-line observations acquired c. 2003. With stand-alone positional uncertainties of ~100 m (DInSAR) and ~1500 m (TerraSAR-X), we estimate combined DInSAR-DInSAR, DInSAR-TerraSAR-X and TerraSAR-X-TerraSAR-X sensor-based epochal uncertainties to be 141, 1503 and 2121 m, respectively (Supplementary Fig. 3; Supplementary Data 2). The SAR-derived values of $\Delta GL_{rate}$ (and, by implication, $U_{\Delta GLR}$) presented in Fig. 1 (see also Supplementary Fig. 3; Supplementary Data 2) were derived in the same manner to those associated with our optical-image analyses.

## Ice-flow acceleration
To supplement our observations of grounding-line change along the West Antarctic coastline, we calculated the change in acceleration in ice-surface flow (m yr[−2]) between c. 2003–2008 and c. 2010–2015

(Fig. 1) using several ice-velocity mosaics obtained from SAR- and optical-based imaging techniques[8,51–53]. A list of the velocity mosaics used in our analyses is included in Supplementary Table 1.

For the year 2015, we used the NASA ITS_LIVE 2015 Annual Antarctic ice-velocity mosaic[8,53]. This product is available free of charge from the NASA ITS_LIVE data archive (https://nsidc.org/apps/itslive/), and offers near-complete spatial coverage of coastal West Antarctica as derived from the error-weighted average of all Landsat 8 image-pair velocity fields that have a centre date falling within 2015[8]. For years 2003, 2008 and 2010, where spatial coverage of ice velocity is more limited, we generated stacked grids of ITS_LIVE[8,53] and MEaSUREs[51,52]-derived velocity mosaics falling within ± 2 years (for c. 2008 and c. 2010) and ± 4 years (for c. 2003) of our grounding line observations (Supplementary Fig. 4; Supplementary Table 1). These stacks represent the per-pixel median ice-velocity magnitude derived from all input datasets (Supplementary Fig. 4) which, together with the 2015 ITS_LIVE velocity mosaic, were initially decimated onto a common (1×1 km) resolution grid prior to stacking.

Final grids of epochal acceleration between c. 2003-2008 and c. 2010-2015 were calculated by differencing the earlier year of each epoch from the latter, and using temporal baselines of 11.5 and 8 years, respectively. These baselines reflect the temporal span of all input velocity mosaics used in the creation of each grid (Supplementary Fig. 4j; Supplementary Table 1), and thus offer conservative estimates of epochal acceleration and its change through time. Resulting grids were then differenced to obtain the between-epoch change in ice-flow acceleration shown in Fig. 1.

Uncertainties in our velocity and acceleration grids (Supplementary Fig. 4) were quantified as follows: for each year in which we generated stacked velocity records (i.e. c. 2003, c. 2008, c. 2010), we first calculated the median standard error ($U_{vel}$) associated with each input dataset utilised (Supplementary Fig. 4; Supplementary Table 1). Median standard error was calculated to minimise potential bias caused by the ITS_LIVE record, whose standard error values over West Antarctica are typically much greater than those associated with (near-) contemporaneous SAR-based products[51,52]. This is attributed to the effects of, for example, cloud-contamination, snowfall/blow and (pre-Landsat 8) sensor radiometric quality issues inherent in optical-image feature-tracking[8,53], all of which act to limit observation densities leading to potentially biased standard error values. In this instance, formal propagation of biased ITS_LIVE-derived errors with SAR-based estimates would also yield overestimated combined errors relative to the maximum values observed in each input data stack. For these reasons, we double-weighted all SAR-based observations in our calculation of median standard error and, following the metadata associated with the various velocity datasets utilised[8,51–53], consider all derived uncertainty estimates generated in this study to be a sensitive indicator of relative data quality (i.e. the amount of observations in each data stack) rather than absolute error.

Grids of uncertainty associated with our c. 2003-2008 and c. 2010-2015 epochal acceleration observations ($U_{a\_epoch}$) were subsequently derived by summing the median standard error of each year in quadrature (i.e. $\sqrt{\widetilde{U_{vel2003}}^2 + \widetilde{U_{vel2008}}^2}$ and $\sqrt{\widetilde{U_{vel2010}}^2 + \widetilde{U_{vel2015}}^2}$), and were then inputted into a similar calculation to estimate the combined uncertainty, $U_a$, associated with the change in acceleration observations presented in Fig. 1. This combined uncertainty is shown in Supplementary Fig. 4 and, as alluded to above, is considered to represent a conservative, high-end error estimate based upon the propagation of relative errors associated with all input datasets.

## Ice-sheet Mass Change
We examined Antarctic Ice Sheet mass-balance trends derived from Bayesian hierarchical modelling of all existing laser- and radar-altimeter, GRACE and in-situ GPS records acquired over the ice sheet

between 2003 and 2015[54]. Available as annual means for each glacier drainage basin[55], the relative contributions of ice-dynamical and surface processes for each basin draining to the Amundsen Sea (basins 320, 321 and 322) were combined and used to determine total mass-loss trends over the observational period (Supplementary Fig. 8a). Ref. 54 provides a comprehensive discussion of the Bayesian hierarchical modelling approach used to generate these estimates, as well as their uncertainties. Data were acquired free of charge from the 'GlobalMass' project (https://www.globalmass.eu/).

For comparison with the trends outlined above, we additionally examined contemporaneous mass-balance trend estimates as derived from the 2016 Ice sheet Mass Balance Intercomparison Exercise (IMBIE)[1] (Supplementary Fig. 8b). These trends represent annual ensemble mean mass-loss estimates for the entire West Antarctic Ice Sheet (WAIS) between 1992 and 2017, as calculated from numerous independent satellite altimetry-, gravimetry- and mass-budget-based estimates[1]. This dataset was acquired from the IMBIE web portal (http://imbie.org/), and detailed information on the ensemble estimates, calculation techniques and uncertainty quantifications associated with this dataset can be found in ref. 1.

**Atmospheric forcing — Sea-level pressure and Ekman upwelling**
Our mean sea-level pressure (SLP) datasets (Figs. 2a, 2b and 4) were derived from the European Center for Medium-range Weather Forecasts (ECMWF) ERA5 global climate reanalysis[56], which is considered to provide the most accurate depiction of climatic conditions around Antarctica[57,58]. The ERA5 data archive was accessed free of charge at: https://cds.climate.copernicus.eu/. The data shown in Fig. 2a, b were generated using monthly mean of daily mean records, and represent the mean anomaly in SLP between 2003-2008 and 2010-2015, respectively, relative to the period 1979-2015. Mean sea-level pressure anomalies were examined to obtain a first order understanding of the prevailing wind conditions offshore from West Antarctica during each epoch, with importance for associated rates of Ekman upwelling along the coast (see below for further information). The seasonal data shown in Fig. 4 were similarly derived, and represent the difference in mean SLP anomaly (that is, $\overline{SLP}_{2010-2015} - \overline{SLP}_{2003-2008}$) across all complete austral spring (SON), summer (DJF), autumn (MAM) and winter (JJA) cycles spanning 2003-2008 and 2010-2015.

In the absence of high spatial-and temporal-resolution in-situ ocean observations offshore of West Antarctica, we also examined for anomalous changes in Ekman vertical velocity, $wE$ (Fig. 2c and Supplementary Figs. 5 and 6), which approximates the wind-driven upwelling of interior ocean masses, including CDW[18,25]. $wE$ was obtained from[18,59]:

$$wE = \nabla \times \left( (\tau_x, \tau_y) \cdot \frac{1}{\rho_{ocean} f} \right) \quad (1)$$

$$f = 2\omega Sin(\varphi) \quad (2)$$

where $\tau_x, \tau_y$ denote ERA5-derived zonal and meridional wind-stresses (Pa) parameterised to account for the influence of sea ice[60,61], respectively; $\rho_{ocean}$, the assumed density of the Ekman layer (1027.5 kg m$^{-3}$), and $f$ (s$^{-1}$), the Coriolis parameter at latitude $\varphi$ given Earth's angular velocity, $\omega$ (7.292 × 10$^{-5}$ rad s$^{-1}$). In this study, positive $w_E$ denotes Ekman upwelling. Numerous studies have detailed the close correspondence between Ekman upwelling at the continental-shelf break, on-shelf flooding of CDW, and near-contemporaneous glacier change along this and other regions of Antarctica[18,24,25,38,60,62], thereby justifying its use in this study.

Equation 1 does not account for the entire range of complex, local-scale oceanographic processes controlling on-continental shelf CDW transmission towards West Antarctica's coastline[6,28,63–65]; for example, it does not factor in ocean surface current and sea-ice motion-related

interactions that several studies suggest can influence on-shelf Ekman velocities and ocean heat content[6,28,64,65]. Except for the strong near-shore downwelling signals exhibited near the Getz and Dotson ice shelves (cf. Fig. 2c), which mirror those reported in several other studies using more sophisticated Ekman vertical velocity calculations parameterised to account for such on-shore processes[6,64,65], we therefore restrict our discussion of changes in Ekman velocity to those at the continental-shelf break. There, recent work[6,65] has revealed that "accounting for sea ice [and other such ocean-related processes] has little effect on zonal wind stress anomalies at the shelf break"[6], providing confidence in the reliability of the Ekman velocity estimates we report.

As per our SLP analyses, we primarily used ERA5 records to determine $wE$ owing to its robustness in capturing zonal and meridional winds around coastal Antarctica compared with other reanalysis products[57]. We note, however, that the same calculations applied to several independent, coarser resolution reanalyses[66–68] yield the same broad-scale patterns of $wE$ as exhibited in our ERA5-derived estimates, especially with increased distance from the coast where surface winds are well resolved and not influenced by complex coastal topography (Supplementary Fig. 7). This finding suggests that the changes in $wE$ we observe along West Antarctica's continental-shelf break are not biased by, or dependent on, the reanalysis product used. The ERA-Interim, JRA-55 and MERRA-2 reanalysis data used to create Supplementary Fig. 7 are available from https://apps.ecmwf.int/datasets/, http://search.diasjp.net/en/dataset/JRA55 and https://gmao.gsfc.nasa.gov/reanalysis/MERRA-2/, respectively.

**Atmospheric forcing — Southern Annular Mode and El Niño-Southern Oscillation**
The monthly Southern Annular Mode (SAM) index presented in Supplementary Fig. 9a is derived from an array of in-situ-based measurements detailed in ref. 38, and represents the zonal pressure difference as measured between 40°S and 65°S. This dataset is freely available at https://legacy.bas.ac.uk/met/gjma/sam.html.

The monthly El Niño-Southern Oscillation timeseries presented in Supplementary Fig. 9b is that of the 'Oceanic Niño Index' (ONI), and represents the 3-month running average equatorial sea-surface temperature anomaly measured between 5°N and 5°S and 170°W and 120°W from the Extended Reconstructed Sea Surface Temperature Version 4 (ERSST.v5) dataset[69]. This dataset is freely available at: https://psl.noaa.gov/data/correlation/oni.data.

**Oceanic change**
To corroborate our Ekman vertical velocity observations, we also examined corresponding changes in the vertical hydrography of the Southern Ocean fringing West Antarctica using the ensemble median of four independent, eddy-permitting ocean reanalysis products: the ECMWF's Ocean Reanalysis System 5[70] (ORAS5), the UK Met Office's Forecast Ocean Assimilation Model-Global Seasonal Forecasting Service 5 version 13[71,72] (FOAM-GloSea5v13), the Euro-Mediterranean Center on Climate Change's (CMCC) C-GLORS05 reanalysis[73] and Mercator Océan's Global Ocean Reanalysis and Simulations 2 version 4[74] (GLORYS2v4). All products are posted on a common horizontal and vertical grid (0.25° x 0.25° and 75 levels, respectively), and are based on the Nucleus for European Modelling of the Ocean (NEMO) model[75]. Each reanalysis product is also constrained by quality-controlled ocean observations provided by the UK Met Office's EN4 global objective analysis[76] (for ORA5, FOAM-GloSea5v13, C-GLORS05) or the Coriolis Ocean database for Reanalysis[77] (for GLORYS2v4), and are published together as the European Copernicus Marine Environment Monitoring Service's (CMEMS) Global Ocean Ensemble Physics Reanalysis product[78] (GREP). GREP was chosen for use in this study due to its potentially more realistic, ensemble-based representation of the state

of the Southern Ocean at any one time rather than, for example, the use of a single reanalysis product which may be subject to unknown bias. GREP is publicly available at https://marine.copernicus.eu/, and further information on this product, its ensemble reanalysis members and their data assimilation strategies can be found in ref. 78.

To examine for changes in CDW upwelling offshore of West Antarctica, we calculated monthly median ocean potential temperature ($\theta$; °C) and salinity (PSU) across GREP's 75 vertical levels for the period 2003-2015 across all model grid cells on the continental shelf (defined here as <1000 m depth) of the Bellingshausen, Amundsen (encompassing the shelf waters seaward of both the Amundsen Sea Embayment and Getz Ice Shelf regions) and Ross Sea sectors (Fig. 3). Following earlier studies[25,27,28], we identify CDW as having an ocean potential temperature of between 1–2 °C and a minimum salinity of 34.62 PSU.

We acknowledge that all four of the ocean reanalyses products described above are based on similar NEMO configurations which are not parameterised to account for, for example, sub-ice shelf cavity circulation and the presence of icebergs, both of which may induce structural biases[70–75,79–81]. In the Southern Ocean, the observational data assimilated into these reanalyses are also sparse[76,77], but the generally close correspondence between the results we present in Fig. 3 and in-situ observations[25–30] suggests that the reanalyses perform well.

## Data availability

All satellite and climate reanalysis datasets utilised in this study are publicly available and can be obtained from the following sources: https://earthexplorer.usgs.gov/ (Landsat imagery); https://nsidc.org/apps/itslive/ (ITS_LIVE velocity grids); https://nsidc.org/data/ (NASA MEaSUREs velocity grids and grounding lines); https://www.globalmass.eu/ (Bayesian Hierarchical Modelling mass balance outputs); http://imbie.org/ (IMBIE mass balance outputs); https://cds.climate.copernicus.eu/ (Copernicus/ECMWF ERA5 outputs); https://apps.ecmwf.int/datasets/ (ECMWF ERA-Interim outputs); http://search.diasjp.net/en/dataset/JRA55 (Japan Meteorological Agency JRA-55 outputs); https://gmao.gsfc.nasa.gov/reanalysis/MERRA-2/ (NASA Merra2 outputs); https://legacy.bas.ac.uk/met/gjma/sam.html (SAM Index data); https://psl.noaa.gov/data/correlation/oni.data (ERSST.v5 ONI Index data) and https://marine.copernicus.eu/ (CMEMS GREP outputs). The processed grounding-line and velocity data generated in this study have been deposited in the Cambridge Apollo database under accession code https://doi.org/10.17863/CAM.90820 (ref. 82). Supplementary Data 1 contains a list of all satellite products used in the production of our grounding-line datasets, and Supplementary Data 2 contains the calculated grounding-line migration rate values used to produce Fig. 1 and Supplementary Figs. 2 and 3.

## Code availability

The MATLAB codes developed for this study are available on the GitHub page of Frazer Christie at: https://github.com/frazer-christie/Christie_etal_2023_NatComms.

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

## Acknowledgements

This study was supported by a Carnegie Trust for the Universities of Scotland Carnegie PhD Scholarship (to F.D.W.C.), hosted in the Edinburgh E3 U.K. Natural Environment Research Council (NERC) Doctoral Training Partnership (NE/L002558/1) and the Scottish Alliance for Geoscience, Environment and Society (SAGES) Graduate School. The study was also produced with the financial assistance of the Prince Albert II of Monaco Foundation (to F.D.W.C.), as well as grants from the NERC / U.S. National Science Foundation (NSF) International Thwaites Glacier Collaboration (Grants NE/S006613, ITGC-GHOST, to R.G.B. and NE/S006796, ITGC-PROPHET, to N.G.; this is ITGC Contribution No. ITGC-088), NSF Grant 2045075 to E.J.S., NERC Grant NE/T001607/1 (QuORUM) to N.G. and S.F.B.T., and European Space Agency funding (Projects 4000128611/19/I-DT, 4D Antarctica and Digital Twin Antarctica) to N.G. The authors thank I. Joughin for providing access to his TerraSAR-X-derived grounding zone data acquired over Pine Island Glacier (cf. ref. 15).

## Author contributions

F.D.W.C., R.G.B. and E.J.S. conceptualised the study. F.D.W.C. carried out all acquisition and processing of the satellite, climate and ocean reanalysis datasets. F.D.W.C, R.G.B., E.J.S., N.G., and S.F.B.T. contributed to the interpretation of the data. F.D.W.C. wrote the manuscript and produced the figures, with R.G.B., E.J.S., N.G., and S.F.B.T. contributing edits.

## Competing interests

The authors declare no competing interests.
