## [Peer Review File · Nature Communications]

Inter-decadal climate variability induces differential ice response along Pacific-facing West AntarcticaREVIEWER COMMENTS

Reviewer #1 (Remarks to the Author):

The authors assemble diagnostic glaciological variables – grounding line position, ice flow speed, and mass loss – and compare them to atmospheric and oceanic reanalyses to understand the forcings controlling the behavior of the West Antarctic Ice Sheet between 2003 and 2015. The data and methods seem reasonable, and I believe there is a lot of value in the presented observations. Unfortunately, I think the authors missed some important literature that caused them to build their manuscript on an incorrect premise (that marine ice sheet instability – MISI – and response to ocean/atmosphere forcing are mutually exclusive), and that ultimately led them to a conclusion that is in conflict with what their data show. The story that remains is still a valuable scientific contribution, but it is more incremental to literature that has already been published than the claims that are made in the current draft. While the data and methods could largely remain, I think that the analysis in this paper would need to be rethought and rewritten to be publishable. I explain each of my reservations in more detail below.

The study is built on an incorrect premise that relies on mismatched timescales. The second sentence of the abstract states the driving question for the study: “However, the degree to which these losses represent topographically-controlled, runaway retreat divorced from external perturbation, or instead signify an ongoing response to changes in the atmosphere and ocean, is unresolved.” This sets up a dichotomy: either West Antarctica is retreating due to MISI, or it’s responding to atmosphere/ocean changes. This dichotomy is emphasized throughout the paper, as this idea from the abstract is repeated near the beginning of the Introduction (lines 42-45), in the last line of the Introduction (“...we show that the ice sheet response continues to be modulated by subdecadal atmospheric forcing, suggesting that irreversible collapse is not yet certain” lines 54-56) and the first and last paragraph of the Discussion (lines 160-165 and 222-224, respectively). The authors also structure their analysis around proving that West Antarctica responded to ocean/atmosphere forcing during their study period, which they set as a contrast to MISI.

The problem is that MISI operates on a long-term timescale, while ocean/atmosphere forcings may affect the rate of that retreat in the short-term. These are not, therefore, mutually exclusive conditions. Many studies document externally forced variations in retreat rate in dynamically retreating glaciers and ice sheets. For example, Enderlin et al. (2018) document seasonal and sub-seasonal changes in response to geometric and environmental forcings during tidewater glacier retreat. Christianson et al. (2016) document the drop in ocean heat content observed in this study and discuss its impacts on Pine Island Glacier specifically, noting in their abstract:

“...despite a minor, temporary decrease in ice discharge, the basin-wide thinning signal did not change. Thus, as predicted by theory, once marine ice sheet instability is underway, a single transient high-amplitude ocean cooling has only a relatively minor effect on ice flow.” (Christianson et al. 2016)

Seroussi et al. (2017) examine the same phenomenon for all of West Antarctica using a coupled ice/ocean model, and conclude in their abstract “Bed topography controls the pattern of grounding line retreat, while oceanic thermal forcing impacts the rate of grounding line retreat.”

The fact that the rate of grounding line retreat and ice flow acceleration decreased during the study period is interesting, and that sort of variability is important to projecting short-term rates of West Antarctic ice loss due to MISI. However, the motivation of the paper should be reconsidered and the text rewritten to take into account the extensive literature documenting the simultaneous effects of ice-sheet geometry and external forcing on the pace of ice sheet collapse.

Not all examples support the argument. As described above, the paper seeks to establish that West Antarctica responds to ocean/atmosphere forcing, which is evidence that it is not experiencing irreversible MISI. However, not all of West Antarctica is currently in a dynamically unstable

configuration. The manuscript compares the Bellingshausen Sea, where overall rates of retreat accelerated during the study period, with the Amundsen Sea, where overall rates of retreat declined. The declining rate of retreat in the Amundsen Sea was used as evidence for ocean/atmosphere forcing over MISI, but not all of the Amundsen Sea is experiencing MISI. The Getz Ice Shelf, for example, which is used as an example in several places in the manuscript, drains an area of West Antarctica that is primarily well above sea level and not likely to destabilize, even under significant warming (Holschuh et al. 2014). Change in this region is therefore not a good example for understanding ocean/atmosphere forcing and MISI. If the authors want to examine ocean/atmosphere influence on glaciers experiencing MISI, they should at least limit their analysis to catchments with a reverse bed slope.

Since the external forcing/MISI dichotomy is not supported by the literature, the authors might consider presenting a manuscript that quantifies in detail the varying character of the response of Amundsen Sea and West Antarctic-wide ice shelves and glaciers to the decrease in ocean heat content documented during this time period, taking into account glacier geometry. While that would require fairly substantial reworking of the analysis, the comparison between the Amundsen and Bellingshausen Seas as a whole is an interesting start, and more detail could certainly be found in the results presented here.

Presentation of the data is misleading. The paper presents the *rate* of change of grounding line retreat rate and of ice *acceleration* between the time periods. There's nothing wrong with presenting these metrics, but they are easily confused and need to be stated clearly. Line 91 discusses "widespread reductions in ice flow" and 93 says "ice flow reduced by 113%." While the rate of speed increase presumably slowed during that time, an ice flow speed decrease of that magnitude was certainly not observed in the discussed areas (e.g. Christianson et al. 2016). I assume the authors meant to specify that the rate of acceleration was reduced during this time period. While this is likely just a minor typo, it relates to a larger problem: the authors extensively discuss this change in acceleration rate as evidence against MISI, but don't point out that acceleration and grounding line retreat continue during this time period, despite a substantial reduction in ocean heat content, which is evidence that strongly supports MISI.

The originality of the results is overstated. The authors claim in their abstract that they "derive the first comprehensive inter-decadal record of glacier retreat around West Antarctica's Pacific-facing margin." I'm not sure how to define "comprehensive," but it's certainly a claim that's been made before, for example by Rignot and Scheuchl in a 2011 study: "we present 15 years of comprehensive, high-resolution mapping of grounding lines in Antarctica." I also find it an overstatement to define a 12-year study period as "inter-decadal," particularly when it compares two time periods where the nearest data points are just two years apart.

The authors do recognize that their documentation of a change in ocean heat content during this time period has been reported in many other studies, citing several of them. The originality in this paper is in the closer observations of the seasonal patterns of the forcing, and the correlation with varying response across all of West Antarctica. The analysis could certainly be rewritten to highlight these interesting contributions.

Results should not be presented as current. The study period in the manuscript is 2003-2015, which the authors state is the period including the most comprehensive grounding line data availability. While many of their datasets continue through the present, I see no issue with writing a paper that doesn't rely on the most recent data, instead focusing on past patterns. However, throughout the manuscript, the results are presented as representing the current state of West Antarctica, rather than representing a past event. For example, in the abstract: "...the pace, magnitude and extent of ice destabilisation around West Antarctica continues to be modulated by changes in atmospheric circulation." Since the results demonstrate big changes within a 12-year study period, it is dangerous to present the results as though they are current when the most recent data in the study are six years old.

Christianson, K., et al. (2016), Sensitivity of Pine Island Glacier to observed ocean forcing, *Geophys. Res. Lett.*, 43, 10,817– 10,825, doi:10.1002/2016GL070500.

Enderlin, E. M., O'Neel, S., Bartholomaeus, T. C., & Joughin, I. (2018). Evolving environmental and geometric controls on Columbia Glacier's continued retreat. *Journal of Geophysical Research: Earth Surface*, 123, 1528– 1545. <https://doi.org/10.1029/2017JF004541>

Holschuh, N., Pollard, D., Alley, R.B., Anandakrishnan, S. (2014). Evaluating Marie Byrd Land stability using an improved basal topography. *Earth and Planetary Science Letters*, 408, 362-369.

P. Milillo, E. Rignot, P. Rizzoli, B. Scheuchl, J. Mouginot, J. Bueso-Bello, P. Prats-Iraola, Heterogeneous retreat and ice melt of Thwaites Glacier, West Antarctica. *Sci. Adv.* 5, eaau3433 (2019).

Rignot, E., Mouginot, J., and Scheuchl, B. (2011), Antarctic grounding line mapping from differential satellite radar interferometry, *Geophys. Res. Lett.*, 38, L10504, doi:10.1029/2011GL047109.

Seroussi, H., Nakayama, Y., Larour, E., Menemenlis, D., Morlighem, M., Rignot, E., and Khazendar, A. (2017), Continued retreat of Thwaites Glacier, West Antarctica, controlled by bed topography and ocean circulation, *Geophys. Res. Lett.*, 44, 6191– 6199, doi:10.1002/2017GL072910.

Reviewer #2 (Remarks to the Author):

Review of "Inter-decadal climate variability induces differential ice response along Pacific-facing West Antarctica" by Frazer D.W. Christie, Eric J. Steig, Noel Gourmelen, Simon F. B. Tett and Robert G. Bingham.

General comments:

The paper is interesting and it is important to understand contemporary changes before aiming for projections or paleo-reconstructions. All the data analyses seem good (although I am not an expert in remote sensing). The attempts to link to climate drivers are interesting as well, but not entirely convincing. Indeed, the current analysis is a lot based on current understanding of the Amundsen Sea dynamics (importance of Ekman pumping and ENSO) and does not try to challenge or refine this point of view (see my detailed comments). After major revisions along these lines, I think that the manuscript could become suitable for Nature Communications.

1- The part describing the anomalies in Ekman velocities is not very clear:

1a) What part of the Ekman velocity pattern is relevant for CDW intrusions? What are the relevant anomalies for ice-shelf melt? Is it the anomaly at the shelf break near the entrance of the bathymetric troughs, the value near the ice shelf fronts, or the value integrated over the continental shelf, or having all this together? Given that this is not clearly explained, the discussion on Fig. 2c is unclear and the related conclusions do not appear to be well supported.

1b) There are two periods with two different Ekman velocities that may induce different thermocline depths. However, this does not bring robust evidence that change in Ekman velocities is the only difference between the two periods (in other words, correlation does not imply causality). How different is sea ice production over the continental shelf between the two periods (and could that affect the thermocline depth)? I know that a large part of the literature supports the Ekman pumping as a cause of ice-shelf melt, but alternative/complementary hypotheses have not been well tested. I

guess that sea ice production could be diagnosed from the ocean–sea-ice reanalyses. Can it contribute to explain changes in the thermocline depth or can it be rule out as a driver?

2- The authors conclude that the modulation of the ice sheet dynamics by atmosphere variability brings evidence that a marine ice sheet instability (MISI) is not underway, which does not seem correct to me. Both based on a very similar methodology, Joughin et al. (2014) and Favier et al. (2014) concluded that a MISI was underway for Thwaites and Pine Island, respectively. Indeed, when they imposed melt rates higher than present-day values for a few decades, it then required much lower melt rates than at present to go back to the initial grounding line position and ice volume above floatation. However, these authors noted that a reduction in melt reduced the rate of ice mass loss, which is exactly what is described in the present manuscript, but which does not appear to contradict an on-going MISI. Therefore, I am not sure that the finding that external forcing continues to modulate change over inter-decadal timescales can be opposed to the fact that a MISI may already have been triggered. A possible way to see it is that a general trend towards a MISI may be modulated by climate variability. Besides, the present manuscript states that "the bed geometry is conducive of a runaway retreat", but the bed slope is not sufficient to determine stability, as ice-shelf buttressing also matters (Gudmundsson et al. 2012). On the MISI and reversibility of Pine Island's retreat, see also Rosier et al. (2020).

3- The discussion on the role of SAM and ENSO is not based on new analyses and does not appear to be robust. For example, it is claimed that ENSO is very important through its effect of the Amundsen Sea Low (ASL) but Hosking et al. (2013; their Tab. 2) do not report strong correlations between ENSO and ASL indices. The minimum would be to use the ENSO and ASL indices to investigate whether they could explain the changes between the two periods. Having said that, ENSO is very diverse (e.g. Capotondi et al. 2015) and has a substantial inter-decadal variability (e.g. Wittenberg 2009), so I am not sure it makes sense to discuss its impact based on two 6-year periods (or the authors should refer to specific El Niño/La Niña events rather than ENSO as a general mode of variability).

Specific comments:

- L 148-149: Not sure they are independent as they are part of the observational data base assimilated in the ocean reanalyses.

- The methods section should specify that the stress used to calculate the Ekman velocity is the air–sea + air–sea-ice stress (not the stress actually felt by the ocean).

- "Ocean Change" in the Methods section: Using the median of four ocean reanalyses is interesting, but these are all based on similar NEMO configurations with structural biases, e.g. no ice shelf cavity or icebergs, which may induce biases in the Amundsen Sea (e.g., Nakayamma et al. 2014a; Donat-Magnin et al. 2017; Bett et al. 2020). It is true that data assimilation in these reanalyses may compensate for these structural biases, but observational data are quite sparse there and only cover the summer period (e.g., Heywood et al. 2016). These limitations should be mentioned. I nonetheless acknowledge that the fact that "Independent in-situ observations from Pine Island Bay, Dotson and Getz ice shelves corroborate these findings" (L. 148-149) is a very supportive point.

- L 200-201: correlation does not imply causality. At minima, you should explore other possible mechanisms (e.g. changes in sea ice formation) to see if you can rule them out.

Reviewer #3 (Remarks to the Author):

This paper reports on a new dataset of glacier and grounding line retreat in West Antarctica, derives rates of change between two time periods, and relates these to changes in ice flow acceleration over similar periods of time. By showing that the correlated changes between these time periods could result from the effect of changes in atmospheric circulation on warm water inflow to the grounding line, the paper concludes that patterns of atmospheric circulation, rather than ice sheet instability, may be the primary cause of recent ice loss.

This review focusses on the glaciological observations (grounding lines and surface velocities) because of a lack of expertise in the atmospheric and oceanographic methods and datasets employed.

I find all aspects of the literary quality of this paper to be very good and have only a few comments below. The paper is very easy to follow, well grounded in appropriate literature, and the figures are generally both useful and well presented.

With the assumption that the atmospheric and oceanographic analyses are sound, I believe the paper potentially represents a useful contribution to the literature, and recommend publication following some amendments.

GENERAL COMMENTS

The study uses data up to and including 2015. Since this was quite a time ago now, later freely available velocity data exist (2016-2018 at least), and there have been rather significant changes in ice velocity and retreat since 2015, I think the study period need some more justification. Are you sure that studying a wider time period would won't result in a different outcome? Please add some more justifying text.

Given the general availability of datasets for the time period studied, the authors appear to have made appropriate selections of grounding line and velocity data, and reasonable choices as to what time-periods to use for comparing rates of change. Nevertheless, the potential impact of the paper may be undermined by the sometimes wide range of dates on aggregated datasets. This potential weakness is exacerbated by the need to report on changes in ice acceleration, which will have amplified the impact of both the generally increasing quality of data, and the decreasing narrowness of date ranges, with time. I don't think there is much to be done about this, except to be careful about the claims made. On one hand the overall correspondence ("largely associated", line 91) between grounding line retreat rate and ice flow acceleration rate is compelling (and the main reason I support publication). On the other hand, I find the reported changes in the acceleration of Thwaites near the ice divide (line 95) to be highly questionable given the low velocities and high noise in this region (and lack of reported uncertainty). I recommend that the reporting of changes in acceleration be limited only to the regions close (10km-ish) to the grounding line where features for tracking are clear in all datasets and the velocity signal is generally strong and greater than expected uncertainties.

Uncertainties in ground-line observation and differences are addressed. However, these "standalone" values are very difficult to relate to the signals presented because the signals are aggregated over 30km segments. I would like to see an attempt to place the uncertainties on a segment basis so that the significance of the results are easier to judge. Uncertainties in velocities and accelerations are not given, and this is an omission. The change in ice flow acceleration seems compelling, but cannot be properly assessed without some comment on its uncertainty. I know this is difficult, but please at least discuss it.

There is a general sentiment throughout the paper that previous studies have attributed recent ice loss exclusively to the marine ice sheet instability (note line 224), yet I don't believe that to be the case. Whilst I agree that the evidence presented here appears to show that recent trends may be attributed to coupled atmosphere and ocean forcing, suggesting that other studies have ignored the possibility of multiple factors would need a strong reference to back it up. Maybe this sentiment could

be toned down a bit.

SPECIFIC COMMENTS

There appear to be some sections of the paper a little out of place. The second half of the final paragraph of the introduction appears to pre-empt the conclusion, and I suggest that much of it could be removed. Likewise, the end of the first paragraph of the discussion also confronts the reader with the outcome before the evidence is properly reviewed.

Line 594: "input datasets were additionally co-registered". This puzzled me, since all velocity products are already projected. Why was this necessary? How large was the correction? What impact might it have had? Please give further information here.

The figures do a great job of presenting the evidence and are well designed. I have some suggestions for improvements:

- Figure 1: The numbered glacier basins have no key that I could find (without referring to another paper)
- Figure 1: Please make the text labels a bit clearer (I struggled to see the ?s)
- Figure 1 inset: Please use bolder colours
- Figure 2: panels a and b – is there a reason for using contours here? Aliasing on my printer makes these difficult to interpret
- Figure 2: "grey patches" doesn't properly describe the light highlighting of the study periods.

END

Adrian Luckman

"Inter-decadal climate variability induces differential ice response along Pacific-facing West Antarctica" F.D.W. Christie et al.

Point-by-point responses to reviewers

Reviewer #1 (Remarks to the Author):

1. The authors assemble diagnostic glaciological variables – grounding line position, ice flow speed, and mass loss – and compare them to atmospheric and oceanic reanalyses to understand the forcings controlling the behavior of the West Antarctic Ice Sheet between 2003 and 2015. The data and methods seem reasonable, and I believe there is a lot of value in the presented observations. Unfortunately, I think the authors missed some important literature that caused them to build their manuscript on an incorrect premise (that marine ice sheet instability – MISI – and response to ocean/atmosphere forcing are mutually exclusive), and that ultimately led them to a conclusion that is in conflict with what their data show. The story that remains is still a valuable scientific contribution, but it is more incremental to literature that has already been published than the claims that are made in the current draft. While the data and methods could largely remain, I think that the analysis in this paper would need to be rethought and rewritten to be publishable. I explain each of my reservations in more detail below.

We thank the reviewer for their insightful review and the recognition that “*there is a lot of value in the presented observations*”. We are grateful for the pointers to the extra citations. As explained in our response to **Major Comment 1** above, we concur with the reviewers’ guidance on how to phrase the superimposition of the inter-annual variability onto MISI. We now make it clear that the novel contribution of this manuscript lies in the extensive and systematic observations spanning the Pacific-facing WAIS margin compared with previous research which has largely been confined to the Amundsen Sea Sector.

2. The study is built on an incorrect premise that relies on mismatched timescales. The second sentence of the abstract states the driving question for the study: “However, the degree to which these losses represent topographically-controlled, runaway retreat divorced from external perturbation, or instead signify an ongoing response to changes in the atmosphere and ocean, is unresolved.” This sets up a dichotomy: either West Antarctica is retreating due to MISI, or it’s responding to atmosphere/ocean changes. This dichotomy is emphasized throughout the paper, as this idea from the abstract is repeated near the beginning of the Introduction (lines 42-45), in the last line of the Introduction (“...we show that the ice sheet response continues to be modulated by subdecadal atmospheric forcing, suggesting that irreversible collapse is not yet certain” lines 54-56) and the first and last paragraph of the Discussion (lines 160-165 and 222-224, respectively). The authors also structure their analysis around proving that West Antarctica responded to ocean/atmosphere forcing during their study period, which they set as a contrast to MISI.

The problem is that MISI operates on a long-term timescale, while ocean/atmosphere forcings may affect the rate of that retreat in the short-term. These are not, therefore, mutually exclusive conditions. Many studies document externally forced variations in retreat rate in dynamically retreating glaciers and ice sheets. For example, Enderlin et al. (2018) document seasonal and sub-seasonal changes in response to geometric and environmental forcings during tidewater glacier retreat. Christianson et al. (2016) document the drop in ocean heat content observed in this study and discuss its impacts on Pine Island Glacier specifically, noting in their abstract:

“...despite a minor, temporary decrease in ice discharge, the basin-wide thinning signal did not change. Thus, as predicted by theory, once marine ice sheet instability is underway, a single transient high-amplitude ocean cooling has only a relatively minor effect on ice flow.” (Christianson et al. 2016)

Seroussi et al. (2017) examine the same phenomenon for all of West Antarctica using a coupled ice/ocean model, and conclude in their abstract “Bed topography controls the pattern of grounding line retreat, while oceanic thermal forcing impacts the rate of grounding line retreat.”

The fact that the rate of grounding line retreat and ice flow acceleration decreased during the study period is interesting, and that sort of variability is important to projecting short-term rates of West Antarctic ice loss due to MISI. However, the motivation of the paper should be reconsidered and the text rewritten to take into account the extensive literature documenting the simultaneous effects of ice-sheet geometry and external forcing on the pace of ice sheet collapse.

Thank you for this detailed summary which, as we have explained above, has motivated us to change the premise of the paper accordingly. The changes we have implemented are described in our response to **Major Change 1** above, and we have now included references to both Christianson et al. (2016) and Seroussi et al. (2017) in our revised manuscript.

3. Not all examples support the argument. As described above, the paper seeks to establish that West Antarctica responds to ocean/atmosphere forcing, which is evidence that it is not experiencing irreversible MISI. However, not all of West Antarctica is currently in a dynamically unstable configuration. The manuscript compares the Bellingshausen Sea, where overall rates of retreat accelerated during the study period, with the Amundsen Sea, where overall rates of retreat declined. The declining rate of retreat in the Amundsen Sea was used as evidence for ocean/atmosphere forcing over MISI, but not all of the Amundsen Sea is experiencing MISI. The Getz Ice Shelf, for example, which is used as an example in several places in the manuscript, drains an area of West Antarctica that is primarily well above sea level and not likely to destabilize, even under significant warming (Holschuh et al. 2014). Change in this region is therefore not a good example for understanding ocean/atmosphere forcing and MISI. If the authors want to examine ocean/atmosphere influence on glaciers experiencing MISI, they should at least limit their analysis to catchments with a reverse bed slope.

We believe that by “*the argument*”, the reviewer refers to our previous narrative placing atmosphere-ocean forcing in opposition to MISI which we have actioned as per our responses above.

Our main argument in the paper is clarified in the new sentences contained on **Lines 186-191 (Discussion section)** of the revised manuscript:

“Collectively, our observations show that the rate of ice response (grounding-line retreat and ice deceleration) from the Amundsen Sea Embayment to the western Getz Ice Shelf is in-phase with atmosphere-ocean forcing over inter-decadal timescales. The pervasiveness of this relationship along a margin transected by multiple glaciers, with bed slopes ranging from reverse to prograde⁹, signals the primary influence of the inter-decadal forcing on rates of glaciological change”.

We do not argue that bed topography is unimportant; rather, our observations highlight that in spite of topographic context and even along Getz Ice Shelf, the grounding line positions

fluctuated in line with the time scales over which we have noted in the paper ocean-atmosphere forcing also fluctuated. Therefore, we disagree with the sentiment that change in the Getz region is not a good example for understanding atmosphere-ocean forcing, and that there is much to be pursued in this area – but we do agree that, ultimately, regions characterized by reverse bed slopes continue to deserve the most attention.

4. Since the external forcing/MISI dichotomy is not supported by the literature, the authors might consider presenting a manuscript that quantifies in detail the varying character of the response of Amundsen Sea and West Antarctic-wide ice shelves and glaciers to the decrease in ocean heat content documented during this time period, taking into account glacier geometry. While that would require fairly substantial reworking of the analysis, the comparison between the Amundsen and Bellingshausen Seas as a whole is an interesting start, and more detail could certainly be found in the results presented here.

This is a good suggestion. We have reworked the manuscript as described in our responses above.

5. Presentation of the data is misleading. The paper presents the rate of change of grounding line retreat rate and of ice acceleration between the time periods. There's nothing wrong with presenting these metrics, but they are easily confused and need to be stated clearly. Line 91 discusses "widespread reductions in ice flow" and 93 says "ice flow reduced by 113%." While the rate of speed increase presumably slowed during that time, an ice flow speed decrease of that magnitude was certainly not observed in the discussed areas (e.g. Christianson et al. 2016). I assume the authors meant to specify that the rate of acceleration was reduced during this time period. While this is likely just a minor typo, it relates to a larger problem: the authors extensively discuss this change in acceleration rate as evidence against MISI, but don't point out that acceleration and grounding line retreat continue during this time period, despite a substantial reduction in ocean heat content, which is evidence that strongly supports MISI.

We agree that the wording on Lines 91 and 93 of the original manuscript was poorly phrased, and that indeed when ice acceleration reduces (now worded as 'deceleration' in the revised manuscript) this does not mean that the ice is not continuing to flow oceanward as the long-term response to MISI.

Line 82 of the revised manuscript (**Glaciological change section**; formerly Line 91) now reads:

"Whereas localised increases in ice-flow acceleration were detected along parts of the coastline (including, for example, stretches of central Getz Ice Shelf in line with previous research^{8,23}), the observed slowdown in grounding-line retreat was largely associated with ice-flow deceleration, including at and inland of the Pine Island, Thwaites and Smith glacier's grounding lines (Figure 1)".

We have now removed the sentence contained on Line 93 of the original manuscript.

In a similar vein, Line 172 of the revised manuscript (**Discussion section**) now reads:

"This was associated with a deceleration in ice flow along the grounding line of most outlet glaciers (Figure 1) and, by implication, a slowdown in the rate of ice-dynamical imbalance through time".

In addition, in the **Glaciological change section**, we have amended Line 60 to read:

"We also calculated near-contemporaneous change in ice-surface flow acceleration between the two periods ...".

The change of emphasis in the revised manuscript away from MISI addresses the final part of the above comment.

6. The originality of the results is overstated. The authors claim in their abstract that they “derive the first comprehensive inter-decadal record of glacier retreat around West Antarctica’s Pacific-facing margin.” I’m not sure how to define “comprehensive,” but it’s certainly a claim that’s been made before, for example by Rignot and Scheuchl in a 2011 study: “we present 15 years of comprehensive, high-resolution mapping of grounding lines in Antarctica.” I also find it an overstatement to define a 12-year study period as “inter-decadal,” particularly when it compares two time periods where the nearest data points are just two years apart.

As the reviewer highlights, the appropriateness of the word ‘*comprehensive*’ is subjective. In Figure 1 of this rebuttal document, we compare the relative spatial completeness of all MEaSUREs grounding line data (including Rignot et al., 2011; 2014; Scheuchl et al., 2016) derived from InSAR observations acquired post-2000 with the 2015 dataset presented in our paper (see also our Extended Data Figure 1 for the coverage associated with all other years).

We accept that the term ‘inter-decadal’ was a stretch for describing the 12-year period of glaciological observations, but it remains appropriate for describing the atmosphere-ocean forcing elsewhere in the paper including in the title. Consequently, the two places we made changes to the text are as follows:

Line 16 of the **Abstract** now reads:

“Here, we derive a comprehensive, 12-year record of glacier retreat around West Antarctica’s Pacific-facing margin ...”.

The first sentence of the **Figure 1’s caption** now reads:

“Glaciological change across West Antarctica’s Pacific-facing margin, 2003-2015” (rather than *“Inter-decadal glaciological change across West Antarctica’s Pacific-facing margin, 2003-2015”*).

Figure 1 | Spatial coverage of our grounding line observations. *Left panel* shows 2015 grounding line (GL) coverage as presented in the present study (see also Extended Data Figure 1 for coverage associated with all other years) compared with, *right panel*, all MEaSUREs v2 GLs acquired post 2000. Colours represent GLs observed in different years.

7. The authors do recognize that their documentation of a change in ocean heat content during this time period has been reported in many other studies, citing several of them. The originality in this paper is in the closer observations of the seasonal patterns of the forcing, and the correlation with varying response across all of West Antarctica. The analysis could certainly be rewritten to highlight these interesting contributions.

We thank the reviewer for suggesting this alteration to the way we frame the paper, and believe that our revisions to the manuscript satisfy this suggestion. The manuscript now closes highlighting, as the reviewer suggests, the varying response across all of the Pacific-facing WAIS: for example, Line 224 (the penultimate sentence of the Discussion section) now reads:

“Ice margins fringing the Ross Sea changed little reflecting a general barrier to CDW ingress; ice response along the Amundsen Sea was highly sensitive to CDW fluctuations driven by inter-decadal variability in the position of the Amundsen Sea Low; while grounding lines meeting the Bellingshausen Sea retreated progressively in response to pervasive and persistent CDW forcing less susceptible to modulation by the Amundsen Sea Low”.

8. Results should not be presented as current. The study period in the manuscript is 2003-2015, which the authors state is the period including the most comprehensive grounding line data availability. While many of their datasets continue through the present, I see no issue with writing a paper that doesn't rely on the most recent data, instead focusing on

past patterns. However, throughout the manuscript, the results are presented as representing the current state of West Antarctica, rather than representing a past event. For example, in the abstract: "...the pace, magnitude and extent of ice destabilisation around West Antarctica continues to be modulated by changes in atmospheric circulation." Since the results demonstrate big changes within a 12-year study period, it is dangerous to present the results as though they are current when the most recent data in the study are six years old.

We thank the reviewer for raising this point, which we have addressed in our response to **Major Comment 2** above.

Reviewer #2 (Remarks to the Author):

General comments:

The paper is interesting and it is important to understand contemporary changes before aiming for projections or paleo-reconstructions. All the data analyses seem good (although I am not an expert in remote sensing). The attempts to link to climate drivers are interesting as well, but not entirely convincing. Indeed, the current analysis is a lot based on current understanding of the Amundsen Sea dynamics (importance of Ekman pumping and ENSO) and does not try to challenge or refine this point of view (see my detailed comments). After major revisions along these lines, I think that the manuscript could become suitable for Nature Communications.

We thank the reviewer for their constructive review and in particular the guidance that follows in the more specific comments.

1. The part describing the anomalies in Ekman velocities is not very clear:

1a) What part of the Ekman velocity pattern is relevant for CDW intrusions? What are the relevant anomalies for ice-shelf melt? Is it the anomaly at the shelf break near the entrance of the bathymetric troughs, the value near the ice shelf fronts, or the value integrated over the continental shelf, or having all this together? Given that this is not clearly explained, the discussion on Fig. 2c is unclear and the related conclusions do not appear to be well supported.

The anomaly at the shelf break is most important.

We first mention Ekman velocity anomalies in the **Atmospheric Variability section** of the manuscript (**Lines 113-114** of revised document). This refers the reader to the **Methods section**. In the Methods section on Lines 649-652, our original manuscript contained the sentence "*numerous studies have recently detailed the close correspondence between wE upwelling, synchronous oceanographic variability observed from in-situ records, and near-contemporaneous glacier change along this and other regions of Antarctica ...*". We have revised this sentence to make more explicit that it is the anomaly at the continental shelf break that is most important for both CDW intrusions and ice-shelf melt. New sentence (**Lines 735-737**) reads:

"Numerous studies have detailed the close correspondence between wE upwelling at the continental shelf break, on-shelf flooding of CDW, and near-contemporaneous glacier change along this and other regions of Antarctica^{18,24-25,38,63-64}, thereby justifying its use in this study".

We have also added a clarifying sentence to the caption of our figure showing Ekman velocity anomalies (Fig. 2c):

“Note the change towards reduced upwelling (negative wE) near the entrances of the Pine Island East, West and Siple Troughs (blue triangles)”.

We have also accordingly annotated onto the figure the key locations referred to in the text where the anomalies are of most importance for CDW intrusions and ice shelf melt (blue triangles).

We have also edited Lines 119-121 of the Atmospheric variability section to make it clear that the minor change we observe east of the Amundsen Sea Embayment refers to wE at the continental shelf break of the Bellingshausen Sector, and not the continental shelf as a whole. Revised sentence reads:

“East of the Amundsen Sea Embayment, Ekman velocity changed little near the continental shelf break of the Bellingshausen Sea, from high rates around the entrance of the deep continental-shelf-bisecting Belgica Trough (Figure 2c and Extended Data Figures 5-7)”.

In a similar vein, we have also edited Lines 160-167 of the Discussion to excise mention of on-shelf upwelling, which we appreciate may have been confusing. Sentences now read:

“This stretch of coastline corresponds to where the ice margin overlies the CDW-laden tributaries of the deeply-incised Belgica Trough^{12,19}, which lies in direct contact with the Antarctic Circumpolar Current at the continental-shelf break^{12,26-27}. The ice-atmosphere-ocean coupling is maximised in Eltanin Bay, where ice residing immediately upstream of Belgica Trough’s central trunk exhibited the largest increase in grounding-line retreat (Figures 1, 2c and Extended Data Figure 5). This indicates conditions in the Bellingshausen Sea conducive both to sustained transmission of CDW from the continental-shelf break and its steady delivery to the ice margin”.

1b) There are two periods with two different Ekman velocities that may induce different thermocline depths. However, this does not bring robust evidence that change in Ekman velocities is the only difference between the two periods (in other words, correlation does not imply causality). How different is sea ice production over the continental shelf between the two periods (and could that affect the thermocline depth)? I know that a large part of the literature supports the Ekman pumping as a cause of ice-shelf melt, but alternative/complementary hypotheses have not been well tested. I guess that sea ice production could be diagnosed from the ocean–sea-ice reanalyses. Can it contribute to explain changes in the thermocline depth or can it be rule out as a driver?

We thank the reviewer for allowing us the opportunity to clarify that, in addition to Ekman-induced CDW upwelling, there are other factors at play. We agree, as has been noted elsewhere (e.g. Holland et al., 2019) and briefly mentioned on Line 647 of our original submission, that local-scale oceanographic processes (we had in our minds sea-ice production) can affect thermocline depths on the continental shelf. The challenge is that empirically grounded datasets that would facilitate a deep examination of this issue are sparse.

The reviewer’s comment motivated us to assess the relative sea-ice production over 2003-2008 and 2010-2015, as can be inferred from the satellite passive microwave record (e.g. Fetterer et al., 2022). This analysis has revealed only minor overall changes (+/- 5%) across the continental shelf spanning the Bellingshausen/Amundsen/Ross Seas. We have added this analysis to the paper’s Methods Section (Lines 738-746) and in the new Extended Data Figure 9: new text reads:

“Equation 1 does not account for the entire range of complex, local-scale oceanographic processes controlling CDW transmission towards West Antarctica’s coastline^{6,28,65-66}, for example, it does not factor in sea-ice production-related ice-ocean-atmosphere interactions that limited observations suggest may affect on-continental shelf oceanic heat content²⁸. Extended Data Figure 9, however, reveals limited change in continental shelf sea ice concentration⁶³ (a proxy for sea-ice production) over our study domain between 2003-2008 and 2010-2015. It therefore does not draw clear attention to any significant changes in sea-ice production that coincide with the changes to the upper CDW limit that are shown between 2003 and 2015 in Figure 3’.

For completeness, we also include the new **Extended Data Figure 9** as Figure 2 below.

Figure 2 (Extended Data Figure 9 in revised paper) | Sea ice conditions offshore from Pacific-facing West Antarctica, 2003-2015. a-b, mean sea ice concentration (SIC) over the periods a, 2003-2008 and b, 2010-2015 derived from satellite passive microwave observations⁶³. c, change in SIC between the two periods. d, linear trend in SIC across all months spanning January 2003 to January 2015, inclusive (n= 143). Cyan stippling denotes statistically significant trends at $p < .05$, as determined by a two-tailed Pearson’s Linear Correlation Coefficient test. Statistically significant trends at Pine Island Bay (PIB) are related to the migration of large icebergs through time⁸²⁻⁸³ and should be interpreted with caution. In c and d, warm colours denote increased SIC through time. Note that the colour bar limits of c

and d have been deliberately set to emphasise on-continental shelf sea-ice change. In all panels, dark grey denotes no data.

2. The authors conclude that the modulation of the ice sheet dynamics by atmosphere variability brings evidence that a marine ice sheet instability (MISI) is not underway, which does not seem correct to me. Both based on a very similar methodology, Joughin et al. (2014) and Favier et al. (2014) concluded that a MISI was underway for Thwaites and Pine Island, respectively. Indeed, when they imposed melt rates higher than present-day values for a few decades, it then required much lower melt rates than at present to go back to the initial grounding line position and ice volume above floatation. However, these authors noted that a reduction in melt reduced the rate of ice mass loss, which is exactly what is described in the present manuscript, but which does not appear to contradict an on-going MISI. Therefore, I am not sure that the finding that external forcing continues to modulate change over inter-decadal timescales can be opposed to the fact that a MISI may already have been triggered. A possible way to see it is that a general trend towards a MISI may be modulated by climate variability. Besides, the present manuscript states that "the bed geometry is conducive of a runaway retreat", but the bed slope is not sufficient to determine stability, as ice-shelf buttressing also matters (Gudmundsson et al. 2012). On the MISI and reversibility of Pine Island's retreat, see also Rosier et al. (2020).

This point was rightly raised by the other two reviewers and we have addressed it in our response to **Major Change 1** above. The reviewer's phrasing "*A possible way to see it is that a general trend towards a MISI may be modulated by climate variability*" was very helpful, thank you.

3. The discussion on the role of SAM and ENSO is not based on new analyses and does not appear to be robust. For example, it is claimed that ENSO is very important through its effect of the Amundsen Sea Low (ASL) but Hosking et al. (2013; their Tab. 2) do not report strong correlations between ENSO and ASL indices. The minimum would be to use the ENSO and ASL indices to investigate whether they could explain the changes between the two periods. Having said that, ENSO is very diverse (e.g. Capotondi et al. 2015) and has a substantial inter-decadal variability (e.g. Wittenberg 2009), so I am not sure it makes sense to discuss its impact based on two 6-year periods (or the authors should refer to specific El Niño/La Niña events rather than ENSO as a general mode of variability).

Our paper *is* presenting *new* analyses of SAM/ENSO albeit using the established technique of interrogating reanalysis data (cf. Thoma et al., 2008; Steig et al., 2012; Dutrieux et al., 2014; Walker & Gardner, 2017; Holland et al., 2019). Novelty is provided in the selection of time periods and geographical extent presented, and the intercomparison with our new glaciological database.

In terms of the robustness of our application of the reanalysis data and choices made in identifying the importance of ENSO vs. SAM in moderating ASL, we stand by the methodology as presented in the original submission. It builds upon the findings presented by, for example, Steig et al. (2012), Turner et al. (2017), Jenkins et al. (2017) and Holland et al. (2019) cited in our original submission, which underline the dominant role ENSO plays in influencing the position and depth of the ASL. In turn, this influences changes in the offshore wind field at the continental shelf break, driving CDW upwelling. This concept was nicely articulated by Paolo et al. (2018; *Nature Geoscience*, p. 121):

"Variations in the ASL position and strength are driven by tropical Pacific ocean–atmosphere variability (ENSO) and fluctuations in Southern Hemisphere pressure. ASL central pressure

tends to be lower during positive SAM conditions and La Niña years, and higher during El Niño years”.

Therefore, while we appreciate the reviewer’s sentiment that further exploration of the various individual climatic indices or more examination of individual events could have some validity, very similar exercises performed in previous work have already demonstrated that reanalyses adequately capture the principal climatic controls.

Specific comments:

4. L 148-149: Not sure they are independent as they are part of the observational data base assimilated in the ocean reanalyses.

We have removed the word ‘*independent*’ from this sentence (now **Line 143**).

5. The methods section should specify that the stress used to calculate the Ekman velocity is the air–sea + air–sea-ice stress (not the stress actually felt by the ocean).

For the original submission we did not parameterize the effects of sea ice on wind stress because earlier research has shown that this does not influence rates of Ekman upwelling at the continental shelf break (see, for example, Holland et al., 2019). Prompted by this referee comment, however, for the new submission we have for completeness included this parameterization, as added to **Lines 731-732** of the **Methods Section**. The new **Figure 2c** and **Extended Data Figures 6 and 7** are updated to show these updated calculations, but have not changed the overall patterns of anomalous Ekman upwelling presented.

6. "Ocean Change" in the Methods section: Using the median of four ocean reanalyses is interesting, but these are all based on similar NEMO configurations with structural biases, e.g. no ice shelf cavity or icebergs, which may induce biases in the Amundsen Sea (e.g., Nakayamma et al. 2014a; Donat-Magnin et al. 2017; Bett et al. 2020). It is true that data assimilation in these reanalyses may compensate for these structural biases, but observational data are quite sparse there and only cover the summer period (e.g., Heywood et al. 2016). These limitations should be mentioned. I nonetheless acknowledge that the fact that "Independent in-situ observations from Pine Island Bay, Dotson and Getz ice shelves corroborate these findings" (L. 148-149) is a very supportive point.

We thank the reviewer for raising these points which we have now added to the **Methods Section** on **Lines 783-788**.

7. L 200-201: correlation does not imply causality. At minima, you should explore other possible mechanisms (e.g. changes in sea ice formation) to see if you can rule them out.

Thanks. We have addressed this point in our response to Comment #1b above.

Reviewer #3 (Remarks to the Author):

1. This paper reports on a new dataset of glacier and grounding line retreat in West Antarctica, derives rates of change between two time periods, and relates these to changes in ice flow acceleration over similar periods of time. By showing that the correlated changes between these time periods could result from the effect of changes in atmospheric circulation on warm water inflow to the grounding line, the paper concludes that patterns of atmospheric circulation, rather than ice sheet instability, may be the primary cause of recent ice loss.

This review focusses on the glaciological observations (grounding lines and surface velocities) because of a lack of expertise in the atmospheric and oceanographic methods and datasets employed.

I find all aspects of the literary quality of this paper to be very good and have only a few comments below. The paper is very easy to follow, well grounded in appropriate literature, and the figures are generally both useful and well presented.

With the assumption that the atmospheric and oceanographic analyses are sound, I believe the paper potentially represents a useful contribution to the literature, and recommend publication following some amendments.

We thank the reviewer, Prof. Adrian Luckman, for his positive review and helpful suggestions for minor amendments.

GENERAL COMMENTS

2. The study uses data up to and including 2015. Since this was quite a time ago now, later freely available velocity data exist (2016-2018 at least), and there have been rather significant changes in ice velocity and retreat since 2015, I think the study period need some more justification. Are you sure that studying a wider time period would won't result in a different outcome? Please add some more justifying text.

We have addressed part of this comment in our response to **Major Change 2** above.

In particular, to make clearer the choice of our study period, we have written a new paragraph to open the **Methods section (Lines 574-581)**. Paragraph reads:

“An analysis of glaciological, atmospheric and oceanic changes along the entire Pacific-facing margin of West Antarctica must be underpinned by systematic records. Prior to this study, we generated systematic records of grounding line positions between 2003 and 2015 along West Antarctica’s Marie Byrd Land coast using optical satellite-based techniques^{18,47}, hence our focus on this time-period. Other studies have mapped different proxies for the grounding line since 2015 using independent techniques including altimetry and machine learning applied to InSAR observations⁴⁸⁻⁴⁹. However, reconciling “grounding line” positions identified by different techniques is problematic, hence we chose not to extend our study period beyond 2015”.

Naturally, we are ultimately interested in follow-up work that will continue to expand the time over which we can explore the ice-ocean-atmosphere interactions further – indeed some of the authors are presently supervising a Ph.D. project on this topic. We are nevertheless already confident from the analyses we have done for this paper that the outcome – i.e. as expressed by the paper title *“Inter-decadal climate variability induces differential ice response along Pacific-facing West Antarctica”* – stands.

3. Given the general availability of datasets for the time period studied, the authors appear to have made appropriate selections of grounding line and velocity data, and reasonable choices as to what time-periods to use for comparing rates of change. Nevertheless, the potential impact of the paper may be undermined by the sometimes wide range of dates on aggregated datasets. This potential weakness is exacerbated by the need to report on changes in ice acceleration, which will have amplified the impact of both the generally increasing quality of data, and the decreasing narrowness of date ranges, with time. I don't think there is much to be done about this, except to be careful about the claims made. On one hand the overall correspondence ("largely associated", line 91) between grounding line retreat rate and ice flow acceleration rate is compelling (and the main reason I support publication). On the other hand, I find the reported changes in the acceleration of Thwaites near the ice divide (line 95) to be highly questionable given the low velocities and high noise in this region (and lack of reported uncertainty). I recommend that the reporting of changes in acceleration be limited only to the regions close (10km-ish) to the grounding line where features for tracking are clear in all datasets and the velocity signal is generally strong and greater than expected uncertainties.

We thank the reviewer for raising the need for caution when interpreting the acceleration signals shown in Figure 1 and, in summary, have excised the sentence on Thwaites' ice divide behavior from the main text. As the reviewer recommended, all other reporting pertains to acceleration close to the grounding line (where uncertainties are low; see also our response to Comment #4 below), so no further modification to the text is required. We have, however, edited Line 183/184 of the original manuscript (now **Lines 172-174**) to clarify that we refer to ice-flow acceleration near the grounding line. Sentence now reads:

"... This was associated with a deceleration in ice flow along the grounding line of most outlet glaciers (Figure 1) and, by implication, a slowdown in the rate of ice-dynamical imbalance through time".

On the lack of reported uncertainties, please see our response to Comment #4 below.

4. Uncertainties in ground-line observation and differences are addressed. However, these "standalone" values are very difficult to relate to the signals presented because the signals are aggregated over 30km segments. I would like to see an attempt to place the uncertainties on a segment basis so that the significance of the results are easier to judge. Uncertainties in velocities and accelerations are not given, and this is an omission. The change in ice flow acceleration seems compelling, but cannot be properly assessed without some comment on its uncertainty. I know this is difficult, but please at least discuss it.

Grounding line uncertainties

Beginning with grounding line uncertainties being presented on a segment basis, this is something which we had originally planned to include in the accompanying grounding line dataset upon publication (see Acknowledgments), hence its original omission here. Following the reviewer's suggestion, however, we now also include this as the new Extended Data Figure 3 and wish to reiterate that these values are also contained in Extended Dataset 1.

Velocity uncertainties

Regarding the lack of uncertainty information associated with our velocity/acceleration grids, the reviewer is correct to point out that this was an omission. We have addressed this comment

by producing comprehensive maps of uncertainty associated with *a*), the velocity stack of each year analysed (i.e. *c.* 2003, *c.* 2008, *c.* 2010 and 2015), *b*), the epochal acceleration between successive years (i.e. *c.* 2003-2008 and *c.* 2010-2015) and *c*), the overall change in acceleration between epochs shown in Figure 1. Uncertainties associated with *a*) and *c*) are shown in Extended Data Figure 4 and, for concision, we will include those associated with *b*) in the accompanying dataset alongside all other grids upon publication (see Acknowledgements). We have also now included a detailed description of how these uncertainty values were calculated in the **Methods Section (Lines 664-688)**.

In keeping with the reviewer's Comment #3 above, we also note that our calculated uncertainties match the spatial distribution expected by the reviewer, insofar as that combined standard errors – which we consider to be sensitive indicators of overall data quality rather than absolute error (cf. **revised Methods section**) – are typically lowest near the grounding line where the greatest density of observations exist.

5. There is a general sentiment throughout the paper that previous studies have attributed recent ice loss exclusively to the marine ice sheet instability (note line 224), yet I don't believe that to be the case. Whilst I agree that the evidence presented here appears to show that recent trends may be attributed to coupled atmosphere and ocean forcing, suggesting that other studies have ignored the possibility of multiple factors would need a strong reference to back it up. Maybe this sentiment could be toned down a bit.

This point was rightly raised by the other two reviewers and we have addressed it in our response to **Major Change 1** above.

SPECIFIC COMMENTS

6. There appear to be some sections of the paper a little out of place. The second half of the final paragraph of the introduction appears to pre-empt the conclusion, and I suggest that much of it could be removed. Likewise, the end of the first paragraph of the discussion also confronts the reader with the outcome before the evidence is properly reviewed.

We believe we have addressed the two structural issues identified here in the revisions made in response to **Major Change 1**.

7. Line 594: "input datasets were additionally co-registered". This puzzled me, since all velocity products are already projected. Why was this necessary? How large was the correction? What impact might it have had? Please give further information here.

We have now removed this sentence from the text. Upon calculating the uncertainty values associated with our velocity observations (cf. Comments #3 and #4), we realised that our processing chain inadvertently applied a small shift in the MEASUREs datasets at the geocoding (NetCDF > GeoTIFF) stage. We have now fixed this, however results remain the same since the offsets were systematic.

8. The figures do a great job of presenting the evidence and are well designed.

We thank the reviewer for his kind comments regarding the design of our figures.

I have some suggestions for improvements:

Figure 1: The numbered glacier basins have no key that I could find (without referring to another paper)

We have added to **Figure 1's caption** the following clarification:

“Note that the glacial basins shown are from MEaSURES⁴⁴ but for ease of reference we have numbered them 1-33 from east to west.”

9. Figure 1: Please make the text labels a bit clearer (I struggled to see the ?s)

We have enlarged and emboldened the labels.

10. Figure 1 inset: Please use bolder colours

We have now used bolder colours, and changed them to be more in keeping with those used in the main figure (i.e reds/blues). We have also implemented the same edit to the revised Extended Data Figures 2 and 3.

11. Figure 2: panels a and b – is there a reason for using contours here? Aliasing on my printer makes these difficult to interpret

We have followed the convention of displaying atmospheric pressure gradients in isobaric format. We would therefore prefer to retain the current forms as is, but we would reconsider if the editor/reviewers of the revised manuscript pressed this point.

12. Figure 3: “grey patches” doesn’t properly describe the light highlighting of the study periods.

The caption states “*translucent grey patches*” which we would prefer to retain.

Other minor edits made to the manuscript not directly related to the comments of Reviewers 1-3.

- i. In light of Reviewer 2’s comments, we have removed from Extended Data Figure 5 the map of coastal wE accompanying the observations of I_b change between 2003 and 2008, which we realise may have been confusing with regards to the primary importance of CDW upwelling being at the continental shelf break.
- ii. We have made several small edits to Extended Data Table 1 to provide more information on the exact products used (formatting meant this information appeared cropped out of original Table when in PDF format) and their measurement technique(s).
- iii. The revisions made to the manuscript have necessitated the renumbering of all in-text citations, whose total now equals 83 (up from 69 in the initial version of the manuscript).
- iv. We have added a Data Availability statement following the **Methods Section** in accordance with the Editor’s instructions.

REVIEWER COMMENTS

Reviewer #1 (Remarks to the Author):

The revised version of this manuscript now combines glaciological observations of ice-flow speed and grounding-line migration with data detailing atmospheric and oceanic forcing to reveal differences in response between the Amundsen and Bellingshausen Seas between 2003 and 2015. The authors have made substantial revisions to the presentation of their manuscript, and I in doing so have, I believe, added significant context and value to the presentation of their interesting data. Barring two comments below that I believe are important for clarification in the manuscript, I recommend publication.

Comment 1: I raised concerns previously about terminology that implied that ice flow was slowing down when instead the authors meant that ice flow was still accelerating, but the acceleration was occurring more slowly. The authors responded by changing to the terminology "ice-flow deceleration." I don't think this clarifies the issue. Elsewhere in the manuscript, "acceleration" is used to mean that ice flow is speeding up. It would therefore be assumed, consistent with common usage, that "deceleration" would mean that ice flow is slowing down. I realize that it's a clunky sentence construction, but I don't see any way around using something similar to "decrease in the pace of ice-flow acceleration." There are several places in the manuscript where it would be helpful to edit this terminology.

Comment 2: Figure 1 is jam-packed with information, which I think is necessary in this case, but I also think that means that it's best to leave out any information that's not absolutely necessary to the figure. I spent quite a while trying to figure out the meaning of the question marks. I think the caption is saying that these question marks represent a region with grounding line advance or retreat that has been observed but not in the current study period, but I'm not sure whether that's from times before or after the study period. Perhaps more importantly, I'm not sure why it's relevant; given that the paper seeks to present a comprehensive assessment of grounding-line change without focus on case studies, representing data from other studies only available in a few areas doesn't seem useful. And unless I've just missed it, it does not seem to be discussed in any detail in the text. I suggest removing these from the figure and the caption.

Reviewer #2 (Remarks to the Author):

Second Review of “Inter-decadal climate variability induces differential ice response along Pacific-facing West Antarctica” by Frazer Christie, Eric Steig, Noel Gourmelen, Simon Tett and Robert Bingham.

I still think that this manuscript could become suitable for *Nature Communications*, but I recommend returning the manuscript for an additional round of major revisions. The authors have addressed my concern about the wrong claim that a MISI may not be underway (my 2nd initial major comment). They have partly addressed my 1st major comment in which I asked to clarify the role of Ekman pumping, but there are still several aspects that require clarification, including (i) the exact value of Ekman velocity anomalies at the entrance of the various troughs, (ii) the method used to calculate the Ekman velocity, and (iii) the possible role of sea ice production (or lack thereof). Finally, my 3rd initial comment on ENSO was also not addressed, and I explain below why I do not think it is correct to conclude that this study brings “evidence that the wider Amundsen Sea glacier margin strongly feels the effects of ENSO-induced changes in the Amundsen Sea Low”. My comments are detailed below.

Major comments:

“Atmospheric Variability” section (and corresponding part in the Method section): I thank the authors for better showing where Ekman pumping matters but some details remain unclear. The negative anomaly in Ekman velocity is indeed strong at the entrance of Pine Island Trough East, and this remains valid across the different reanalyses (Ext. Data Fig. 7). However, the anomaly at the entrance of Pine Island Trough West is only weakly negative, as is the anomaly at the entrance of Belgica Trough in the Bellingshausen Sea (Ext. Data Fig. 7). It should also be acknowledged that the signal at the entrance of the Getz-Dotson Trough is slightly positive in ERA5 and not robust across reanalyses (Ext. Data Fig. 7). All this should be described more accurately.

Method section – Atmospheric forcing: To address my comment on the effect of sea ice on the Ekman velocity, the authors have added that the wind stress used in the calculation of the Ekman velocity is “parameterised to account for the presence of sea ice”. This just means that the drag coefficient used in the stress calculation is a combination of the air-sea drag and the air-ice drag (equation 3 of Greene et al. 2017 which is cited for that). But this was not my point. My point is that the actual surface stress felt by the ocean surface (and therefore relevant for upwelling) is a combination of the air-sea stress (for open water) and sea-ice–ocean stress (for water covered by sea ice). The latter depends on the friction related to sea-ice motions and/or surface currents beneath the ice. Dotto et al. (2019, <https://doi.org/10.1175/JPO-D-19-0064.1>) showed the differences in terms of Ekman velocity (see figure below). The fact that this is not considered should be mentioned as a significant caveat.

I am also not at all convinced by the paragraph that has been added on the sea ice production in the same part of the method section. The authors use the mean sea ice concentration on the continental shelf as a proxy for sea ice production. This is a very bad proxy that would, for example, indicate near-zero sea ice production in coastal polynya that are known to have high production rates. On the Amundsen continental shelf, the sea ice volume (concentration times thickness) results from the balance between the net sea ice production and the offshore sea ice advection. As winds change between the two periods, there could be a significant change in the offshore export of sea ice that could balance a significant change in sea ice production for a nearly unchanged sea ice concentration (not even considering possible changes in thickness). The paragraph that has been added (and Ext. Data Fig. 9) should therefore be removed. My initial suggestion was to consider sea ice production as represented in the ocean reanalyses that are used in this paper (this is a classical output from ocean–sea ice models). If nothing is added about sea ice production, then it should not be claimed that the results “clearly implicate wind-driven oceanic change as the major forcing mechanism” (L. 195) (as I wrote in my initial review: correlation does not imply causality).

Discussion section – I was not fully convinced by the response of my comment related to ENSO. Towards the end of their manuscript, the author conclude: “Our findings therefore provide evidence that the wider Amundsen Sea glacier margin strongly feels the effects of ENSO-induced changes in the Amundsen Sea Low, embodied by fluctuations in dynamic thinning” (L. 211-213). While previous studies indeed showed some correlation between ENSO indices and the wind stress near Pine Island East Trough, the fact that the wind-induced Ekman pumping differs between 2003-2008 and 2010-2015 does not bring evidence that ENSO has explained these differences. The Amundsen Sea Low has its own dynamics, which is mostly unrelated to ENSO (Hosking et al. 2013), and changes between the two periods could a priori be unrelated to ENSO. I therefore suggest either removing the claims that ENSO has induced these changes or, at least, showing the differences in ENSO activity between 2003-2008 and 2010-2015.

Other comments:

L. 133 & 139: not sure what “empirical” means for observations.

L. 143-145: About “In-situ observations from Pine Island Bay, Dotson and Getz ice shelves corroborate these findings”: this is not so clear for Dotson, as Jenkins et al. (2018) report high thermocline until 2011 (their Fig. 2) while this study reports low thermocline from approximately 2008. Similarly, Dutrieux et al. (2014) and Webber et al. (2017) reported high thermocline until 2010 for Pine Island (their Fig. 2 and 3, respectively). These disagreements between observed profiles and the ocean reanalyses should be mentioned.

"Inter-decadal climate variability induces differential ice response along Pacific-facing West Antarctica" by F.D.W. Christie et al.

Response to Reviewers, 30/09/2022

We thank Reviewers 1 and 2 for re-reviewing our manuscript and for their further constructive comments. We are pleased to read that the reviewers are now satisfied with our revisions regarding Major Change 1 (MIS1 versus ocean-forcing) and 2 (Description of study timeframe) and that, in the words of Reviewer 1, these changes have "*added significant context and value to the presentation of [our] interesting data*".

As per the format of our initial response document, we provide point-by-point responses to each of the reviewers' additional comments below. Reviewers' comments are numbered in blue and our responses are in black. In our responses we refer in red to the section and line numbers of the revised text with changes implemented (document ending "*_clean_version*"). We also provide a separate document with changes from the previous submission tracked throughout (document ending "*_tracked_changes*").

Point-by-point responses to reviewers

Reviewer #1 (Remarks to the Author):

The revised version of this manuscript now combines glaciological observations of ice-flow speed and grounding-line migration with data detailing atmospheric and oceanic forcing to reveal differences in response between the Amundsen and Bellingshausen Seas between 2003 and 2015. The authors have made substantial revisions to the presentation of their manuscript, and [in] doing so have, I believe, added significant context and value to the presentation of their interesting data. Barring two comments below that I believe are important for clarification in the manuscript, I recommend publication.

We are pleased to read that the reviewer is now satisfied with the presentation of our manuscript and thank them for remarking upon the "*significant context and value to the presentation of [our] interesting data*" these revisions have added. Below, we address the reviewer's two remaining comments in full.

1. Comment 1: I raised concerns previously about terminology that implied that ice flow was slowing down when instead the authors meant that ice flow was still accelerating, but the acceleration was occurring more slowly. The authors responded by changing to the terminology "ice-flow deceleration." I don't think this clarifies the issue. Elsewhere in the manuscript, "acceleration" is used to mean that ice flow is speeding up. It would therefore be assumed, consistent with common usage, that "deceleration" would mean that ice flow is slowing down. I realize that it's a clunky sentence construction, but I don't see any way around using something similar to "decrease in the pace of ice-flow acceleration." There are several places in the manuscript where it would be helpful to edit this terminology.

From a purely physics-based perspective, 'deceleration' is a permissible term to explain this phenomenon since the motion vector (in this case seaward-flowing ice considered on a 2D plane) is in opposite sign to the (negative) acceleration vector and, importantly, the direction of motion does not change between observations. The term 'deceleration' in this context therefore relates to the retardation of velocity over the course of the two periods to some non-zero value (i.e., the ice temporarily slows, but does not cessate).

We appreciate, however, that ambiguity often exists regarding the precise definition of ‘deceleration’ versus, for example, ‘negative acceleration’, so we have opted to adhere to the reviewer’s choice of wording to avoid confusion when discussing ice-flow change.

Lines 82-86 now read: “Whereas localised increases in ice-flow acceleration were detected along parts of the coastline (including, for example, stretches of central Getz Ice Shelf in line with previous research^{8,23}), the observed slowdown in grounding-line retreat was largely associated with a decrease in the pace of ice-flow acceleration, including at and inland of the Pine Island, Thwaites and Smith glacier’s grounding lines (Figure 1).”.

Lines 176-179 now reads: “This was associated with a reduction in the pace of ice flow acceleration along the grounding line of most outlet glaciers (Figure 1) and, by implication, a slowing in the rate of ice-dynamical imbalance through time”.

Lines 184-187 now read: “Changes in near-shore upwelling also explain the net slowdown in grounding-line migration rate and reduction in the pace of ice-flow acceleration at Dotson Ice Shelf (Figure 1 and Extended Data Figure 2), which reduced ice-shelf-averaged thinning²² and facilitated a 2-km re-advance of Kohler Glacier from c. 2011^{13,29}.”.

Lines 188-192 now read: “Similar mechanisms reduced upwelling and limited CDW ingress beneath Getz Ice Shelf⁹⁰, consistent with the net reduction in grounding-line retreat rate we observe along this stretch of the coastline (Figure 1). Collectively, our observations show that the rate of ice response (grounding-line retreat and ice-flow acceleration change) from the Amundsen Sea Embayment to the western Getz Ice Shelf is in-phase with atmosphere-ocean forcing ...”

The caption of **Figure 1** has also been revised to read: “... Data are superimposed over near-contemporaneous change in ice-flow acceleration ($m\ yr^{-2}$) (Methods) ...”.

2. Comment 2: Figure 1 is jam-packed with information, which I think is necessary in this case, but I also think that means that it's best to leave out any information that's not absolutely necessary to the figure. I spent quite a while trying to figure out the meaning of the question marks. I think the caption is saying that these question marks represent a region with grounding line advance or retreat that has been observed but not in the current study period, but I'm not sure whether that's from times before or after the study period. Perhaps more importantly, I'm not sure why it's relevant; given that the paper seeks to present a comprehensive assessment of grounding-line change without focus on case studies, representing data from other studies only available in a few areas doesn't seem useful. And unless I've just missed it, it does not seem to be discussed in any detail in the text. I suggest removing these from the figure and the caption.

We thank the reviewer for this helpful suggestion. Question marks and associated caption have been removed.

Reviewer #2 (Remarks to the Author):

General comments:

I still think that this manuscript could become suitable for Nature Communications, but I recommend returning the manuscript for an additional round of major revisions. The authors have addressed my concern about the wrong claim that a MISI may not be underway (my 2nd initial major comment). They have partly addressed my 1st major comment in which I asked to clarify the role of Ekman pumping, but there are still several aspects that require clarification, including (i) the exact value of Ekman velocity anomalies at the entrance of the various troughs, (ii) the method used to calculate the Ekman velocity, and (iii) the possible role of sea ice production (or lack thereof). Finally, my 3rd initial comment on ENSO was also not addressed, and I explain below why I do not think it is correct to conclude that this study brings “evidence that the wider Amundsen Sea glacial margin strongly feels the effects of ENSO-induced changes in the Amundsen Sea Low”. My comments are detailed below.

We are pleased that the reviewer is now satisfied with our revisions pertaining to MISI and thank them for their additional comments regarding Ekman upwelling- and ENSO-induced change. Most of these comments mirror those expressed below, which we have now addressed in full.

Regarding ‘(ii)’, we were a little surprised to read this comment on the basis that this is explicitly detailed in *Atmospheric Forcing* section of the *Methods*. With the exception of compensation for sea ice- and ocean current-motion (which is arguably relevant only for on-continental shelf Ekman process studies – see our responses below), our technique is otherwise identical to that reported in, for example, Dotto et al. (2019).

Major comments:

1. “Atmospheric Variability” section (and corresponding part in the Method section): I thank the authors for better showing where Ekman pumping matters but some details remain unclear. The negative anomaly in Ekman velocity is indeed strong at the entrance of Pine Island Trough East, and this remains valid across the different reanalyses (Ext. Data Fig. 7). However, the anomaly at the entrance of Pine Island Trough West is only weakly negative, as is the anomaly at the entrance of Belgica Trough in the Bellingshausen Sea (Ext. Data Fig. 7). It should also be acknowledged that the signal at the entrance of the Getz-Dotson Trough is slightly positive in ERA5 and not robust across reanalyses (Ext. Data Fig. 7). All this should be described more accurately.

We thank the reviewer for these comments which we have taken on board. Regarding Pine Island Trough East & West, we have reworked slightly the paragraph beginning **Line 110** to read:

“... No region-wide CDW records exist spanning 2003-2015, but Ekman vertical velocity (as calculated from a variety of global reanalyses products; Methods) reduced significantly (implying less upwelling) near the entrance of Pine Island Trough East (~-5 m yr⁻¹; Figure 2c and Extended Data Figures 6 and 7), the primary routeway of CDW to the Amundsen Sea Embayment^{4,9,24-25}. More muted reductions in Ekman Vertical Velocity (~-1 m yr⁻¹) also occurred near the mouth of the tributary Pine Island Trough West”.

[Note the inclusion of approximate wE values in brackets (derived from the mean of all reanalysis products in ED Figure 7) which additionally satisfies sub-comment (i) of the reviewer’s **General comment** above].

In a similar vein, we have slightly reworded the following sentence on Belgica Trough. **Lines 121-124** now read:

“East of the Amundsen Sea Embayment, Ekman velocity changed little from the high rates near the continental shelf break of the Bellingshausen Sea, with only minor amounts of reduced upwelling ($\sim 1 \text{ m yr}^{-1}$) detected around the entrance of the deep continental-shelf-bisecting Belgica Trough (Figure 2c and Extended Data Figures 5-7)”.

Regarding the reviewer’s comment on Getz-Dotson Trough, we believe this is a small oversight since all reanalyses show positive wE at that location in ED Figure 7. We instead believe this comment refers to the enhanced downwelling signals observed at the mouth of Siple Trough - a phenomenon uncaptured in MERRA-2 compared with all other reanalysis products. The pervasive coastal upwelling in this reanalysis suggests that MERRA-2 fails to reliably capture surface winds close to the coast (as should be expected of a model of that generation and resolution), so we have edited **Lines #741-747** of the **Methods** to explain this discrepancy (and all other minor differences between models on the continental shelf, for that matter). In doing so, we have also slightly reworded the text to emphasise that it is wE values at the continental shelf break which are most important within the context of this study (cf. our response to the reviewer’s comments #2 and #3 below). Revised sentences read:

“We note, however, that the same calculations applied to several independent, coarser resolution reanalyses⁶⁸⁻⁷⁰ yield the same broad-scale patterns of wE as exhibited in our ERA5-derived estimates, especially with increased distance from the coast where surface winds are well resolved and not influenced by complex coastal topography (Extended Data Figure 7). This finding suggests that the changes in wE we observe along West Antarctica’s continental shelf break are not biased by, or dependent on, the reanalysis product used”.

2. **Method section – Atmospheric forcing:** To address my comment on the effect of sea ice on the Ekman velocity, the authors have added that the wind stress used in the calculation of the Ekman velocity is “parameterised to account for the presence of sea ice”. This just mean that the drag coefficient used in the stress calculation is a combination of the air-sea drag and the air-ice drag (equation 3 of Greene et al. 2017 which is cited for that). But this was not my point. My point is that the actual surface stress felt by the ocean surface (and therefore relevant for upwelling) is a combination of the air-sea stress (for open water) and sea-ice–ocean stress (for water covered by sea ice). The latter depends on the friction related to sea-ice motions and/or surface currents beneath the ice. Dotto et al. (2019, <https://doi.org/10.1175/JPO-D-19-0064.1>) showed the differences in terms of Ekman velocity (see figure below). The fact that this is not considered should be mentioned as a significant caveat.

Fig. 1 | Reviewer 2's figure adapted from Dotto et al. (2019), showing the difference between wE parameterized for the effects of wind-driven sea ice / ocean drift vs. wind alone.

We thank the reviewer for raising the importance of sea ice and surface current motion, and acknowledge that our calculations of wE do not account for these processes. In the context of the present study, however, we believe that such parameterisations are an unnecessary complication, and thus respectfully disagree that their omission represents a 'significant caveat'. We outline our reasoning behind this conclusion as follows.

- A) Recent work (e.g., Holland, Steig et al., *Nature Geoscience*, 2019) has shown that in the Amundsen and Bellingshausen Seas, accounting for sea ice and surface motion has little effect on zonal wind stresses (and, by implication, Ekman pumping) at and seaward of the continental shelf break. There, 'wind-only' (i.e., that approximately equivalent to the calculations presented in this study, albeit without the use of sea ice-dependent drag) vs. 'total-stress' (i.e., those parametrized to account for ocean surface and sea ice motion) stresses are closely correlated ($R^2 = 0.65-0.8$). This is nicely articulated on p.725 of their paper when they state: "*Accounting for sea ice has little effect on zonal stress anomalies at the shelf break, so zonal winds [alone] are used as a proxy for zonal stress at the shelf break throughout the study ...*". This finding is also clearly shown in their Supplementary Figure S1a, which we reproduce here as Figure 2 of this rebuttal document. Given these findings, also note the close similarities – as should consequently be expected – between Fig 1a and 1b above at and beyond the continental shelf break; there, Ekman pumping rates are in fact very similar.

Fig 2 | Vector correlation between total stress and wind-only stress, calculated using monthly anomalies from seasonal climatology spanning 1992-2016. Figure adapted from Holland et al. (2019).

B) This and many other studies (including, for example, refs. 3,4,5,6,9,12,18,19,20,21,23,24,25,26,27,28,29,30,31,38,41,48,62,64 and 66 cited in text) have underscored the importance of continental-shelf-bisecting troughs for permitting on-shelf CDW ingress and flooding (see, for example, the differences in hydrography over the Amundsen and (trough-free) Ross Sea regions in Figure 3 of our manuscript). It therefore stands to reason that Ekman processes at the continental shelf break are the first-order control on subsequent rates of ocean-driven glacial melting/retreat, as has indeed been shown in a plethora of earlier studies (see, for example, the observed, close correlation between upwelling and glacial melting as discussed in refs. 5,6,9,10,20,21,23,24,25 and 29, especially).

By extension, therefore, while calculations like those presented in Fig. 1 reveal the undoubtedly complex local-scale Ekman processes operating on-shelf (insofar as surface current patterns are realistic, noting here that beyond models, few long-term observational datasets exist to verify these patterns), these processes can arguably be considered as ones of second-order importance for net ice-shelf melting (cf. refs. 5, 6, 21, 24). This is because, as noted in the *Methods* and elsewhere⁶, such processes tend to reflect the mechanisms controlling the non-linear/non-laminar flow of CDW that is *already on the continental shelf*. In other words, if no CDW ingress occurred at the shelf break in the first place, then significantly less ice-shelf melting would most certainly occur irrespective of these on-shelf processes.

C) It is for these reasons that we were careful to refer only to changes in Ekman velocity at/near the continental shelf break in our initial and revised manuscript. The only exception to this was in our discussion of the enhanced downwelling at the front of the Getz and Dotson ice shelves. Notably, however, this phenomenon corresponds with the location of the Amundsen Sea Polynya which (somewhat unsurprisingly for a polynya), also exhibits downwelling in both Fig. 1 (cf. Dotto et al., 2019) and the sea-ice—ocean-parameterised observations presented by Holland et al. (2019). We therefore think the discussion of this exception in the main text is acceptable, and would prefer to retain it as is.

Notwithstanding the above points, we appreciate that much of the detail presented in A-C could have been more explicitly articulated in the Methods. For these reasons, we have revised **Lines 752-763** as follows:

“Equation 1 does not account for the entire range of complex, local-scale oceanographic processes controlling on-continental shelf CDW transmission towards West Antarctica’s coastline^{6,28,66-67}, for example, it does not factor in ocean surface current and sea-ice motion-related interactions that several studies suggest can influence on-shelf Ekman velocities and ocean heat content^{6,28,67}. Except for the strong near-shore downwelling signals exhibited near the Getz and Dotson ice shelves (cf. Figure 2c), which mirror those reported in several other studies using more sophisticated Ekman vertical velocity calculations parameterised to account for such on-shore processes^{6,67}, we therefore restrict our discussion of changes in Ekman velocity to those at the continental shelf break. There, recent work has shown that “accounting for sea ice [and other such ocean-related processes] has little effect on zonal wind stress anomalies at the shelf break”⁶, providing confidence in the reliability of the Ekman velocity estimates we report”.

3. I am also not at all convinced by the paragraph that has been added on the sea ice production in the same part of the method section. The authors use the mean sea ice concentration on the continental shelf as a proxy for sea ice production. This is a very bad

proxy that would, for example, indicate near-zero sea ice production in coastal polynya that are known to have high production rates. On the Amundsen continental shelf, the sea ice volume (concentration times thickness) results from the balance between the net sea ice production and the offshore sea ice advection. As winds change between the two periods, there could be a significant change in the offshore export of sea ice that could balance a significant change in sea ice production for a nearly unchanged sea ice concentration (not even considering possible changes in thickness). The paragraph that has been added (and Ext. Data Fig. 9) should therefore be removed. My initial suggestion was to consider sea ice production as represented in the ocean reanalyses that are used in this paper (this is a classical output from ocean–sea ice models). If nothing is added about sea ice production, then it should not be claimed that the results “clearly implicate wind-driven oceanic change as the major forcing mechanism” (L. 195) (as I wrote in my initial review: correlation does not imply causality).

We thank the reviewer for this comment which is a direct continuation of the one above. In brief, we agree that the use of SIC as a proxy for production is probably a sub-optimal one, and have removed the figure as suggested. We have also now excised from the text any reference to SIC as a proxy for production (i.e., Lines 734-738 of the initial revised manuscript) as well as the original refs. 82 and 83.

Following our response to comment #2 above, we wish to reiterate here that the incorporation of sea ice production into our wE calculations is unneeded given that our observations, unless otherwise explicitly stated in the text, are restricted to those occurring at the continental shelf break. Any dedicated efforts to incorporate this phenomenon would therefore have little overall impact upon the wE values already presented. We also note that GREP does not contain any sea ice production outputs; ascertaining time-variable sea ice production would therefore likely require the use of high resolution coupled ice-ocean-atmosphere model(s) (like that used by Dotto et al., 2019) which would a), fall greatly outside the remit of this study and, b), be difficult to verify given a dearth of long-term in-situ observations (cf. ref. 6 for further discussion on this point).

On the basis of the above, we believe the inclusion of L195 (now L197) remains justified in the text, and we would therefore prefer to retain it as is. The inclusion of this line moreover makes the findings of our study align with, for example, the large range of pre-existing literature showing the intimate correlation between offshore winds and glacial change in the Amundsen Embayment (cf. main text and our response ‘B’ to comment #2 above).

4. Discussion section – I was not fully convinced by the response of my comment related to ENSO. Towards the end of their manuscript, the author conclude: “Our findings therefore provide evidence that the wider Amundsen Sea glacier margin strongly feels the effects of ENSO-induced changes in the Amundsen Sea Low, embodied by fluctuations in dynamic thinning” (L. 211-213).

While previous studies indeed showed some correlation between ENSO indices and the wind stress near Pine Island East Trough, the fact that the wind-induced Ekman pumping differs between 2003-2008 and 2010-2015 does not bring evidence that ENSO has explained these differences. The Amundsen Sea Low has its own dynamics, which is mostly unrelated to ENSO (Hosking et al. 2013), and changes between the two periods could a priori be unrelated to ENSO.

I therefore suggest either removing the claims that ENSO has induced these changes or, at least, showing the differences in ENSO activity between 2003-2008 and 2010-2015.

Regarding ENSO vs. ASL variability, we were somewhat surprised to read this comment on the basis that in our initial response document, we made explicit reference to a selection of more recent studies which underline the key role ENSO plays in influencing the depth of the ASL. This included an excerpt from a recent paper by Paolo et al. entitled: *Response of Pacific-sector Antarctic ice shelves to the El Niño/Southern Oscillation* (2018; *Nature Geoscience*) that states (p.121):

“Variations in the ASL position and strength are driven by tropical Pacific ocean–atmosphere variability (ENSO) and fluctuations in Southern Hemisphere pressure. ASL central pressure tends to be lower during positive SAM conditions and La Niña years, and higher during El Niño years”.

In addition, our response also referred to the recent review paper by Turner, Bingham et al. (*Reviews of Geophysics*, 2017) — the first author of which was involved directly in the work by Hosking et al. (2013) — who report (p.256):

“...The ASL is therefore strongly influenced by tropical climate variability associated with the El Niño–Southern Oscillation (ENSO) ... In addition, the depth of the ASL is affected by the Southern Annular Mode (SAM) ...”

Ultimately, these studies supersede the work of Hosking et al. (2013), who in any case caveated the apparent lack of correlation between SAM, ENSO and corresponding ASL strength they found by concluding: *“The controlling influence of [ASL] relative central pressure is less marked, although this is thought to be due to variability in longitude masking its impact”* (p.6645). We therefore believe our conclusion on Lines 211-213 is acceptable in the context of this newer research, and would prefer to keep it as is.

That said, the reviewer’s comment prompted us to realize that there were perhaps several logic gaps and/or instances of assumed knowledge in our discussion of the ASL and, specifically, its links to SAM and ENSO forcing. We further realize that discussion of SAM’s long-term trends (or lack thereof) was probably also confusing within the context of our observations spanning 2003-2015, and that knowledge of its change across this time would be more meaningful (particularly in relation to, for example, the excerpt from Paolo et al. (2018) quoted above).

To remedy these issues, improve manuscript clarity and better align the text with the findings of the more recent studies detailed above, we have therefore implemented the following revisions:

- a) With reference to the recent studies discussed above, we now provide additional information on **Lines 205-212** about how SAM likely influenced the observed deepening in SLP / ASL strength (cf. Figures 2c and 4 of the manuscript). Prompted by the reviewer’s suggestion, for completeness, we also now include (as Extended Data Figure 9a; reproduced below as Figure 3a) timeseries of the SAM index between 2003 and 2015.
- b) We have excised mention of SAM’s long-term trend from the text, and instead now report upon the spatial signature of the SLP anomalies shown in Figure 4 as a means in which to link the discussion to ENSO forcing.
- c) We also now discuss (as a new paragraph beginning **Line 215**) the close correlation between ENSO forcing and the observed deepening of the ASL, as well as the tight synchronicity between ENSO and our glaciological observations in the Amundsen Sector. Following the reviewer’s suggestion, the latter is also now shown in the new Extended Data Figure 9b (reproduced below as Figure 3b).

Collectively, this revised section (Lines 197-235) now reads:

“... The close correspondence between glacier behaviour and atmosphere-ocean variability we observe (Figures 1-3) clearly implicates wind-driven oceanic influence as the major forcing mechanism. Notably, the Amundsen Sector lay within ~1000 km from the predominant deepening pressure centre of the Amundsen Sea Low during our observational period, equivalent to the synoptic-scale length of atmospheric deformation at high latitudes³⁵; while most of the Bellingshausen Sector lay beyond (Figure 2b). Several studies have implicated the depletion of stratospheric ozone and related changes to the Southern Annular Mode (SAM), an index for the poleward contraction and strength of the westerly winds encircling Antarctica, in the deepening of the Amundsen Sea Low³⁶⁻³⁷. In general, when SAM is in positive polarity (SAM+), sea-level pressure in the region of the Amundsen Sea Low is lower²⁴. Near-uninterrupted and increasingly positive SAM+ conditions between 2003 and 2015 (Extended Data Figure 9, Methods) may therefore partly explain the intensification of the Amundsen Sea Low we find between 2003 and 2015 (Figure 2b), although, apart from austral summertime, seasonal partitioning of this phenomenon (Figure 4; Methods) reveals zonally asymmetric SLP anomalies around Antarctica uncharacteristic of SAM's typical spatial signature^{5,38}. These seasonal patterns suggest that an additional large-scale forcing mechanism, bolstered by ongoing SAM+ conditions, dominated West Antarctic ice-ocean-atmosphere interaction over the timescales we consider.

Beyond SAM, modelling of atmospheric and oceanic forcing in the Amundsen Sector has linked the wind-driven incursion of CDW to far-field perturbations in atmospheric circulation, with the principal teleconnection being the El Niño-Southern Oscillation (ENSO)^{5-6,25,39-40}. Consistent with our seasonal SLP observations (Figure 4), this teleconnection is most prevalent during non-summertime months^{5,40}. Over our period of observation, a distinct transition from relatively sustained El Niño- to strong, prolonged La Niña-like conditions from c.2010 (Extended Data Figure 9) can therefore explain the anomalous deepening of the Amundsen Sea Low we detect (Figures 2b and 4; cf. refs. 21 and 24). Critically, this transition was also synchronous with the timing of reduced grounding-line retreat rates and ice-flow acceleration we observe along the Amundsen Sector (Figures 1 and Extended Data Figure 9). Our findings therefore provide evidence that the wider Amundsen Sea glacier margin strongly feels the effects of ENSO-dominated changes in the Amundsen Sea Low, embodied by fluctuations in dynamic thinning. In contrast, the Bellingshausen Sea is less affected by such atmospheric variability, owing to both its relative remoteness from the Amundsen Sea Low and the more pervasive flow of CDW onto the continental shelf in this region (Figures 2 and 3). Our observations therefore build upon previous research largely limited to the Amundsen Sea Embayment that has also shown an intimate coupling between inter-decadal atmosphere-ocean variability and ice-sheet change^{21,24-25,29,41}. They suggest that in fact the coupling is particularly strong in the Amundsen Sea Embayment compared with elsewhere. This aligns with a previous assertion deduced from coarse-resolution modelling that interannual ocean variability in the Bellingshausen Sea is weaker than in the neighbouring Amundsen Sea⁴².

d) Finally, we now include a short paragraph in the **Methods section** (under the new heading **'Atmospheric forcing – Southern Annular Mode and El Niño-Southern Oscillation'**) which details the SAM and ENSO indices used to produce the new Extended Data Figure 9. New section (beginning **Line 776**) reads:

“The monthly Southern Annular Mode (SAM) index presented in Extended Data Figure 9a is derived from an array of in-situ-based measurements detailed in ref. 38, and represents the zonal pressure difference as measured between 40°S and 65°S. This dataset is freely available at <https://legacy.bas.ac.uk/met/gjma/sam.html>.

The monthly El Niño-Southern Oscillation timeseries presented in Extended Data Figure 9b is that of the ‘Oceanic Niño Index’ (ONI), and represents the 3-month running average equatorial sea-surface temperature anomaly measured between 5°N and 5°S and 170°W and 120°W from the Extended Reconstructed Sea Surface Temperature Version 4 (ERSST.v5) dataset⁷¹. This dataset is freely available at: <https://psl.noaa.gov/data/correlation/oni.data>”.

Figure 3 (Extended Data Figure 9 of revised manuscript) | Southern Annular Mode and El Niño-Southern Oscillation forcing. Figure shows changes (12-month running means) in the polarity of **a**, the Southern Annular Mode (SAM) and **b**, the El Niño-Southern Oscillation (ENSO) between 2003 and 2015 (Methods). Positive ENSO denotes El Niño-like conditions; negative, La Niña. Translucent red and blue patches delimit the periods 2003-2008 and 2010-2015, respectively, which correspond to the timing of our glaciological observations. Note the close synchronicity between our observations of reduced grounding-line retreat and ice-flow acceleration rates in the Amundsen Sector between 2010-2015 (Figure 1) and the transition towards strong SAM+ and La Niña-like conditions c.2010 (see main text for further discussion).

Other comments:

5. L. 133 & 139: not sure what “empirical” means for observations.

‘Empirical’ refers to measurements “based on, concerned with, or verifiable by observation rather than theory or pure logic” (Oxford dictionary), so we would prefer to retain this wording.

6. L. 143-145: About “In-situ observations from Pine Island Bay, Dotson and Getz ice shelves corroborate these findings”: this is not so clear for Dotson, as Jenkins et al. (2018) report high thermocline until 2011 (their Fig. 2) while this study reports low thermocline from approximately 2008. Similarly, Dutrieux et al. (2014) and Webber et al. (2017) reported high thermocline until 2010 for Pine Island (their Fig. 2 and 3, respectively). These disagreements between observed profiles and the ocean reanalyses should be mentioned.

The thank the reviewer for raising the apparent discrepancies between the ocean reanalyses presented and the in-situ-based observations of Dutrieux et al. (2014), Webber et al. (2017) and Jenkins et al. (2018). Ultimately, it is important to bear in mind that reanalysis models will always be subject to some degree of bias, and thus never represent observed conditions in a perfect manner. In the present study, the use of ensemble means and averaging over the entire continental shelf region (Figure 3) will undoubtedly also introduce smoothing and/or temporal biases relative to the local-scale point measurements detailed above. For the purposes of enabling regional comparisons (e.g. Bellingshausen vs. Amundsen), we therefore think these apparent discrepancies are permissible.

Nonetheless, to remedy the reviewer's concern, we have reworded **Lines 146-148** of the revised manuscript as follows, which we believe is a more eloquent way to convey that slight discrepancies exist between the ensemble mean and in-situ observations without having to explicitly discuss/specify each disagreement in turn. Revised text reads:

"In-situ observations revealing similar trends of oceanic cooling from ~2010 at Pine Island Bay, Dotson and Getz ice shelves align closely with these findings^{25,28-30}".

Other minor edits made to the manuscript not directly related to the comments of Reviewers 1 and 2.

- i. Following the addition of the new section "*Atmospheric forcing – Southern Annular Mode and El Nino-Southern Oscillation*" in the *Methods* (cf. our response to Reviewer 2's comment 4), we have subsequently renamed the original section entitled "*Atmospheric forcing*" to "*Atmospheric forcing – Sea-level pressure and Ekman upwelling*".

-- END --

REVIEWER COMMENTS

Reviewer #2 (Remarks to the Author):

I thank the authors for their responses. I have a few remaining comments that should be easy to address if the authors are not too reluctant.

- I am surprised that the authors were surprised by my comment on the method used to calculate the Ekman velocity given that they just added that “zonal and meridional wind-stresses (Nm s^{-1}) [were] parameterised to account for the influence of sea ice” with a simple reference to two papers. The 2nd modified version explicitly stating that sea ice motions and ocean currents are not considered is slightly better, except that the wind stress units are wrong (τ should be in N m^{-2}). I nonetheless suggest pointing to Figure 2 of Dotto et al. (2019) which shows that the wind stress is a reasonable (not very good) approximation of the stress actually felt by the ocean surface at the shelf break (which is not true further south). In response to the three points used to claim that their approach to calculate Ekman velocities is very good in their responses (I don't expect a response):

A. The work by Holland et al. (2019) shows that there is ~ 0.70 co-variance between the wind stress and the stress received by the ocean at the shelf break, but this was obtained by neglecting ocean surface currents that are quite strong at the shelf break (leading to important ocean–sea-ice relative velocities).

B. If “it stands to reason that Ekman processes at the continental shelf break are the first-order control on subsequent rates of ocean-driven glacial melting/retreat”, and that it was already shown in 25 references, then half of this article, including figures 2 and 3, is useless. Besides, several of these papers point to the importance of wind stress without explicitly showing that the important part is the Ekman upwelling at the shelf break (horizontal Ekman transport and consequences on the eastward undercurrent may be another aspect).

C. Downwelling in a polynya is unsurprising if it is due to convection, not when Ekman vertical velocities are considered.

- About the use of the term “empirical observations” (L. 136, 142, 800), I thank the authors for the quote from the Oxford Dictionary from which I understand that it means “oceanographic observations that are verifiable by observation rather than by theory or pure logic”. Isn't it a tautology?

- The changes made to address Reviewer #1's concern about the use of “acceleration” are generally correct (as far as I noticed), except in Fig. 1, where “change in ice flow acceleration (m.yr^{-2})” (both in the caption and in the figure) should be “ice flow acceleration (m.yr^{-2})” (or possibly something like “ice flow velocity changing rate”).

- L. 148: “closely” should be removed, as a three-year lag is not so close (see 2nd other comment in my previous review).

- Caption of Fig. 4: is “anomalous” really needed? “Difference” is probably enough.

- About “Our findings therefore provide evidence that the wider Amundsen Sea glacier margin is strongly affected by ENSO-dominated changes in the Amundsen Sea Low” (L. 225-226): The authors show that their results are consistent with the role of ENSO that was described in previous studies, which is an important result, but they shouldn't claim that their study provides evidence that the changes are strongly affected by ENSO-dominated changes. Indeed, this conclusion is based on only 2 values of ENSO (2003-2008 vs 2010-2015), which does not appear particularly robust.

- L. 228: the Amundsen Sea Low is not remote from the Bellingshausen Sea, as clearly shown in Fig. 2. It is actually called the Amundsen-Bellingshausen Sea Low in several studies. The authors should

provide a better explanation for the absence of ENSO effect in the Bellingshausen Sea.

"Inter-decadal climate variability induces differential ice response along Pacific-facing West Antarctica" by F.D.W. Christie et al.

Response to Reviewer #2, 25/11/2022

We thank Reviewer 2 for re-re-viewing our manuscript and are happy to read that they are now content with the paper barring a few minor suggested amendments. As per the format of our initial response document, we provide point-by-point responses to each of the reviewer's comments below. The reviewer's comments are numbered in blue and our responses are in black. In our responses we refer in red to the section and line numbers of the revised text with changes implemented (document ending "_clean"). We also provide a separate document with changes from the previous submission tracked throughout (document ending "_tracked_changes").

Reviewer #2 (Remarks to the Author):

I thank the authors for their responses. I have a few remaining comments that should be easy to address if the authors are not too reluctant.

1. - I am surprised that the authors were surprised by my comment on the method used to calculate the Ekman velocity given that they just added that "zonal and meridional wind-stresses (Nm s^{-1}) [were] parameterised to account for the influence of sea ice" with a simple reference to two papers. The 2nd modified version explicitly stating that sea ice motions and ocean currents are not considered is slightly better, except that the wind stress units are wrong (τ should be in N m^{-2}). I nonetheless suggest pointing to Figure 2 of Dotto et al. (2019) which shows that the wind stress is a reasonable (not very good) approximation of the stress actually felt by the ocean surface at the shelf break (which is not true further south).

We thank the reviewer for these suggestions. The issue pertaining to tau units is a good spot and, for better consistency with e.g. Figures 2 and 4, we have changed this to Pascals (Pa). (noting here that $1 \text{ Pa} = 1 \text{ N m}^{-2}$). The suggested inclusion of Dotto is also a good one, and we have added this as the new ref. 68 in the **Methods**. **Lines 449-460** now read:

"Equation 1 does not account for the entire range of complex, local-scale oceanographic processes controlling on-continental shelf CDW transmission towards West Antarctica's coastline^{6,28,66-68}; for example, it does not factor in ocean surface current and sea-ice motion-related interactions that several studies suggest can influence on-shelf Ekman velocities and ocean heat content^{6,28,67-68}. Except for the strong near-shore downwelling signals exhibited near the Getz and Dotson ice shelves (cf. Figure 2c), which mirror those reported in several other studies using more sophisticated Ekman vertical velocity calculations parameterised to account for such on-shore processes^{6,67-68}, we therefore restrict our discussion of changes in Ekman velocity to those at the continental shelf break. There, recent work^{6,68} has revealed that "accounting for sea ice [and other such ocean-related processes] has little effect on zonal wind stress anomalies at the shelf break"⁶, providing confidence in the reliability of the Ekman velocity estimates we report".

2. ... In response to the three points used to claim that their approach to calculate Ekman velocities is very good in their responses (I don't expect a response):

A. The work by Holland et al. (2019) shows that there is ~ 0.70 co-variance between the wind stress and the stress received by the ocean at the shelf break, but this was obtained by neglecting ocean surface currents that are quite strong at the shelf break (leading to important ocean-sea-ice relative velocities).

We thank the reviewer for this comment. This is correct but, as alluded to in our previous response document and explicitly stated in Holland, Steig et al. (2019), the inclusion of such parameters “are not pursued because their requirement for additional unknown quantities implies they may not lead to a more realistic stress”. In the **Methods** we make no attempt to hide the fact that our Ekman calculations may be relatively straightforward in comparison to the more detailed modelling studies mentioned by the reviewer here and in their previous reviews, but on the basis of Holland et al. (2019) (and, by extension, Fig 2 of Dotto et al., 2019), we believe our choice of methodology is both justified and scientifically robust.

B. If “it stands to reason that Ekman processes at the continental shelf break are the first-order control on subsequent rates of ocean-driven glacial melting/retreat”, and that it was already shown in 25 references, then half of this article, including figures 2 and 3, is useless. Besides, several of these papers point to the importance of wind stress without explicitly showing that the important part is the Ekman upwelling at the shelf break (horizontal Ekman transport and consequences on the eastward undercurrent may be another aspect).

We consider this to be an unfair comment on the basis that almost all of these studies pertain to local-scale observations (i.e., changes in upwelling across only one sub-region of the domain presented in this study), those spanning very different observational windows (sometimes only one or two years), or a combination of the two. Regarding horizontal Ekman transport and undercurrents, **Lines 449-460** of the **Methods** also make it clear that these mechanisms (and possibly more) may be at play, but are not necessarily captured in our calculations and/or outputs.

C. Downwelling in a polynya is unsurprising if it is due to convection, not when Ekman vertical velocities are considered.

Thanks. This comment pertains to our previous rebuttal document and not the manuscript text, so no further action is required.

3. - About the use of the term “empirical observations” (L. 136, 142, 800), I thank the authors for the quote from the Oxford Dictionary from which I understand that it means “oceanographic observations that are verifiable by observation rather than by theory or pure logic”. Isn’t it a tautology?

This wording was used to emphasise the fact that, unlike the models we refer to in the text, these observations reflect measurements collected in-situ. We understand the reviewer’s point, however, and so have removed ‘empirical’ from each of the instances listed above. On **Lines 142** and **147**, ‘empirical observations’ now read ‘*in-situ oceanographic observations*’ and ‘*independent ocean observations*’, respectively, to better reflect this point.

4. - The changes made to address Reviewer #1’s concern about the use of “acceleration” are generally correct (as far as I noticed), except in Fig. 1, where “change in ice flow acceleration (m.yr⁻²)” (both in the caption and in the figure) should be “ice flow acceleration (m.yr⁻²)” (or possibly something like “ice flow velocity changing rate”).

We thank the reviewer for these suggestions but respectfully disagree with both. Figure 1 shows the **change in acceleration** arising from two separate acceleration measurements spanning 2003-2008 and 2010-2015, yielding the first suggestion rephrasing inaccurate (this would only be true if we showed the **change in velocity** spanning the two periods). Similarly, ‘ice flow velocity changing rate’ is equivalent to acceleration (i.e., d^2V/dt^2 , units $m\ yr^{-2}$), whereas we report the **change in acceleration** (with units still $m\ yr^{-2}$ since we do not explicitly

calculate the third derivative of time which would yield the 'jerk' of the ice (units m yr^{-3} ; corresponding to d^3V/dt^3). As such, we would prefer to retain Fig 1's caption and legend as is.

5. - L. 148: "closely" should be removed, as a three-year lag is not so close (see 2nd other comment in my previous review).

Done.

6. - Caption of Fig. 4: is "anomalous" really needed? "Difference" is probably enough.

We wish to retain this since it reflects the difference in anomalous pressure calculated for the two time periods relative to the long term (1979-2015) record.

7. - About "Our findings therefore provide evidence that the wider Amundsen Sea glacier margin is strongly affected by ENSO-dominated changes in the Amundsen Sea Low" (L. 225-226): The authors show that their results are consistent with the role of ENSO that was described in previous studies, which is an important result, but they shouldn't claim that their study provides evidence that the changes are strongly affected by ENSO-dominated changes. Indeed, this conclusion is based on only 2 values of ENSO (2003-2008 vs 2010-2015), which does not appear particularly robust.

Point taken, and we have amended **Lines 231-237** as follows to make it clear that our observations pertain only to the period 2003-2015 (noting here that we have also implemented a similar change to the following sentence (**Line 234+**) on the Bellingshausen Sea). We have also tempered the language used to convey that our findings *suggest* that ENSO was the dominant control. Revised text reads:

"Our findings therefore suggest that the wider Amundsen Sea glacier margin was strongly affected by ENSO-dominated changes in the Amundsen Sea Low between 2003 and 2015, embodied by fluctuations in dynamic thinning. In contrast, the Bellingshausen Sea was less affected by such atmospheric variability, owing to both its relative remoteness from the long-term central pressure location of the Amundsen Sea Low and the more pervasive flow of CDW onto the continental shelf in this region (Figures 2 and 3)".

8. - L. 228: the Amundsen Sea Low is not remote from the Bellingshausen Sea, as clearly shown in Fig. 2. It is actually called the Amundsen-Bellingshausen Sea Low in several studies. The authors should provide a better explanation for the absence of ENSO effect in the Bellingshausen Sea.

Here, we had the ASL's long-term central pressure location in mind (close to which atmospheric deformation will be maximal; cf. Fig 2), although we realise this was poorly articulated on Line 227/8 as the reviewer has correctly picked up on. We have therefore amended this line for greater clarity (new **Lines 234-236**), as seen in our response to comment #7 above.

--END--